# Weight Ensembling Improves Reasoning in Language Models

**Xingyu Dang**[*,1]    **Christina Baek**[*,2]    **Kaiyue Wen**[3]    **Zico Kolter**[2]    **Aditi Raghunathan**[2]
[1]Tsinghua University    [2]Carnegie Mellon University    [3]Stanford University
dangxy20@mails.tsinghua.edu.cn    kbaek@andrew.cmu.edu

## Abstract

We investigate a failure mode that arises during the training of reasoning models, where the diversity of generations begins to collapse, leading to suboptimal test-time scaling. Notably, the Pass@1 rate reliably improves during supervised finetuning (SFT), but Pass@k rapidly deteriorates. Surprisingly, a simple intervention of interpolating the weights of the latest SFT checkpoint with an early checkpoint, otherwise known as WiSE-FT, almost completely recovers Pass@k while also improving Pass@1. The WiSE-FT variant achieves better test-time scaling (Best@k, majority vote) and achieves superior results with less data when tuned further by reinforcement learning. Finally, we find that WiSE-FT provides complementary performance gains that cannot be achieved only through diversity-inducing decoding strategies, like temperature scaling. We formalize a *bias-variance tradeoff* of Pass@k with respect to the expectation and variance of Pass@1 over the test distribution. We find that WiSE-FT can reduce bias and variance simultaneously, while temperature scaling inherently trades off between bias and variance.

## 1 Introduction

Recent advances in large language models (LLMs) have showcased their remarkable ability to perform complex reasoning, yet these successes often hinge on test-time scaling strategies (Lightman et al., 2023; Snell et al., 2024; Wu et al., 2024). In many applications, such as math problems, puzzles, and logical reasoning, LLMs employ a verification framework where it is significantly easier for the model to verify a candidate solution than to generate one from scratch. This distinction has given rise to strategies that sample multiple "reasoning traces" or sequences of reasoning steps during inference, selecting the best final guess through an outcome reward model (ORM) or majority vote. In this setting, an upper bound on the performance a model could achieve is measured by Pass@K, or the probability that at least one out of $K$ independently sampled reasoning traces is correct.

Unfortunately, while the standard training pipeline of supervised finetuning (SFT) followed by reinforcement learning (RL) dependably improves Pass@1 for reasoning, Pass@K tends to drop early into finetuning (Cobbe et al., 2021; Chow et al., 2024a; Chen et al., 2025). This mismatch arises from a symptom of finetuning called diversity collapse, where overtuned models yield less diverse generations. This is detrimental to Pass@K since the model wastes $K$ attempts on only a handful of guesses. In fact, by analyzing the model's error rate i.e., $1 -$ Pass@1, across the test distribution, we derive a **Pass@K bias-variance trade-off**. To improve *expected test* Pass@K, one can either reduce the *bias* which is the expected error rate or how much the model's error rate *varies* across problems. The latter term is connected to diversity — more diversity allows models to hedge and do uniformly well across all test questions. In particular, during SFT, Pass@1 improves (bias ↓) at the cost of diversity collapse (variance ↑).

Surprisingly, common ways of alleviating diversity collapse, such as early stopping at peak Pass@K or decoding with high temperature, *suffer from the reverse trade-off*: diversity improves (variance ↓) at the cost of overall Pass@1 degrading (bias ↑). Consequently, in this paper we are concerned with a central question:

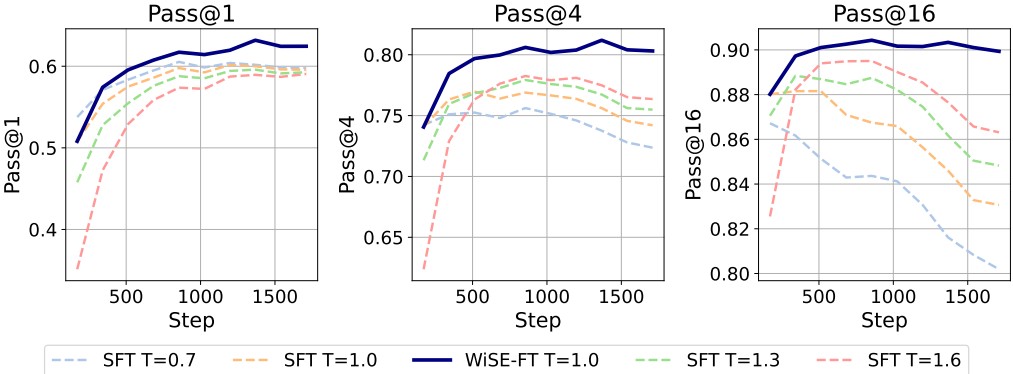

Figure 1: **Pass@k of WiSE-FT versus SFT on GSM8k** Gemma-2-2B supervised finetuned and evaluated on GSM8k. At each SFT timestep $t$, we evaluate Pass@k of checkpoint $\boldsymbol{w}_t$ (in dashed) with its WiSE-FT variant $1/2 \cdot \boldsymbol{w}_t + 1/2 \cdot \boldsymbol{w}_0$ (in solid), where traces are independently sampled with temperature $T = [0.7, 1.0, 1.3, 1.6]$.

*Is it possible to simultaneously improve both* Pass@1 *and* Pass@K, *thereby overcoming the bias-variance tradeoff inherent in current approaches?*

In our work, we introduce a simple, scalable and effective intervention that allows models to achieve both high Pass@K and Pass@1 across mathematical reasoning tasks GSM8k, MATH, and AIME. The specific technique we use is a variant of WiSE-FT (Wortsman et al., 2022) where we interpolate the weights of the latest SFT checkpoint $\boldsymbol{w}_t$ with an early checkpoint $\boldsymbol{w}_0$ as $\boldsymbol{w}_{\mathsf{WiSE}(t)} = 1/2 \cdot \boldsymbol{w}_0 + 1/2 \cdot \boldsymbol{w}_t$. Our key finding is that WiSE-FT successfully merges the diverse sampling capabilities of earlier checkpoints while retaining or surpassing the Pass@1 of later checkpoints. In Figure 1, we observe that the WiSE-FT model achieves *both* higher Pass@K and Pass@1 with more SFT steps $t$, unlike naive SFT which suffers from an early decay in Pass@K. Moreover, the gains with WiSE-FT is unachievable by early-stopping or diversity-aware decoding alone.

Thus, we propose a new paradigm of training reasoning models: 1.) Train extensively using SFT as long as Pass@1 improves, 2.) Perform WiSE-FT with an earlier SFT checkpoint, 3.) Continue tuning the WiSE-FT variant using RL. Overall, the WiSE-FT model has the following immediate practical benefits:

- **Better Test-Time Scaling** Across all datasets and base models, the WiSE-FT variant achieves the highest performance with test-time scaling (Majority Vote, ORM) compared to an overtrained SFT model paired with diversity-aware decoding.

- **Better Reinforcement Learning** Since RL uses self-generated data to tune models, to generalize reliably, it is important for generations to provide sufficient learning signal while also having high coverage over the data space. We find that continued RL training starting from WiSE-FT weights achieves superior results with less synthetic data compared to initializing RL from the last SFT checkpoint and even early-stopped SFT.

In summary, we provide a comprehensive analysis of how reasoning models suffer from diversity collapse during SFT and its negative downstream impact during RL and test-time scaling. We first discuss our WiSE-FT findings in §4. Motivated by this discovery, we investigate two fundamental questions. First, we investigate diversity collapse during SFT and RL of reasoning models in §5. Diversity collapse not only impacts the model's ability to attempt different guesses. In fact, we make an even stronger observation — the generations of reasoning models converge towards *a single reasoning trace* for each test question. We theoretically prove that standard RL algorithms (i.e., REINFORCE and GRPO) fail to recover lost diversity in a simplified discrete bandit setting.

Second, we formalize the competing goals of Pass@1 and Pass@K as a bias-variance tradeoff in §6. We empirically measure and compare the bias and variance of WiSE-FT versus

early-stopping versus high temperature decoding. Notably, only WiSE-FT reduces both bias and variance. We conclude with a remark on the limitations of decoding strategies such as top-k (Shao et al., 2017), nucleus (Holtzman et al., 2020), and min-p (Nguyen et al., 2024), at eliciting the maximum capabilities with test-time scaling from current reasoning models.

## 2 Related Works

**Diversity collapse with SFT:** The standard pipeline for enhancing reasoning in LLMs involves an initial phase of supervised fine-tuning (SFT) followed by reinforcement learning (RL) (Guo et al., 2025; Setlur et al., 2024). SFT is critical for instilling interpretable and readable reasoning chains and ensuring that the model adheres to a consistent rollout templates (Guo et al., 2025). However, a number of recent works have identified critical pitfalls of SFT that hinders the model's ability to explore and ultimately it's overall problem solving ability. Notably, Cobbe et al. (2021) observe diversity collapse when finetuning on GSM8k training dataset, during which the Pass@1 continuously improves whereas Pass@k starts to fall shortly into the training. Similar diversity collapse phenomenon also exists in the self-improvement setting with SFT (Song et al., 2024), and is theoretically investigated as the sharpening effect (Huang et al., 2024). This is not desirable as diverse sampling at inference is important for test-time scaling using majority voting (Wang et al., 2023) or reward model guided search (Setlur et al., 2024; Beeching et al., 2024). Yeo et al. (2025); Chu et al. (2025) attribute this behavior to overfitting, memorization of samples and overfixation to a template style leading to reduced generalization. In our work, we corroborate similar findings and propose ensembling over the course of SFT as a mitigation strategy.

**Mitigating diversity collapse:** Given the importance of diversity for effectively scaling inference-time compute, several recent works have proposed auxiliary finetuning objectives and decoding strategies to mitigate diversity collapse. Li et al. (2025) regularize the SFT process using a game-theoretic framework that encourages sparse updates, thereby preserving output diversity. Zhang et al. (2024b) directly optimizes for diversity during finetuning. Other approaches modify the finetuning procedure to directly optimize for Best-of-N sampling at inference time (Chow et al., 2024b; Sessa et al., 2024; Chen et al., 2025). Another line of work focuses on inference-time decoding, explicitly encouraging diverse solutions through modified beam search strategies (Vijayakumar et al., 2018; Olausson et al., 2024; Chen et al., 2024; Beeching et al., 2024). Li et al. (2023) improve diversity during parallel decoding by appending curated prompts to the input. In formal reasoning settings e.g., Lean, methods such as Monte Carlo tree search have been used to diversify intermediate reasoning steps, as demonstrated in AlphaProof (AlphaProof and AlphaGeometry teams, 2024). In this work, we identify a simple and complementary intervention during the finetuning process to maintain the diversity of generations. We especially care about enforcing diversity while preserving the overall accuracy of generations.

## 3 Preliminaries and Experimental Setup

### 3.1 Pass@k, Best@k, and Majority Vote

Given a reasoning model $f(\cdot)$, a decoding strategy $D$, and problem $x$, the model's solution is obtained by sampling a *reasoning trace* $\boldsymbol{r} := [x, s^{(1)}, s^{(2)}, ..., s^{(n)}, \hat{y}]$ consisting of a sequence of intermediate steps $s^{(i)}$ and a final guess $\hat{y}$. Given $k$ independently sampled traces, Pass@K measures the probability that at least one guess matches the true answer $y$:

$$\text{Pass@K}(x) = \mathbb{E}_{[r_i]_{i=1}^k \sim D(f(x))} \left[ \mathbb{1}\{\exists\, i \in [k] \text{ s.t. } \hat{y}_i = y\} \right] = 1 - (1 - \rho_x)^K \tag{1}$$

where $\rho_x = P(\hat{y} = y \mid x, f, D)$ is the *Pass@1* or marginal probability of sampling the ground truth answer. Then $(1 - \rho_x)^K$ is the probability that all $K$ guesses are incorrect. We will refer to Pass@1 as $\rho_x$ interchangeably in our paper.

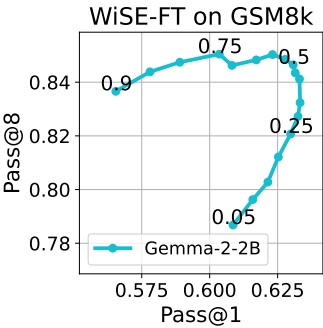 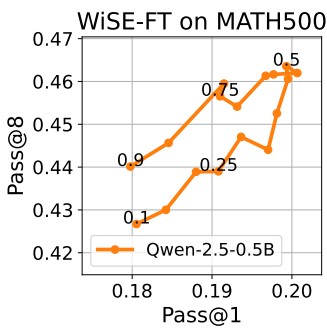 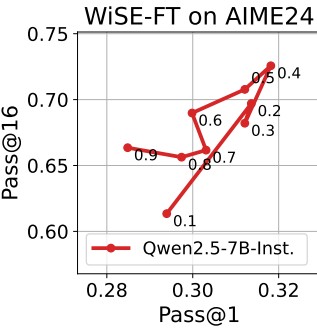

Figure 2: **Pass@1 vs. Pass@K across Interpolation Coefficients** We perform WiSEFT with $\delta \in [0.1, 0.9]$ between the first and last checkpoints of model (in legend) finetuned on GSM8K, MATH, and OpenThoughts-114K, then evaluate on GSM8K, MATH500, and AIME24, respectively. Early SFT model observe higher Pass@K (y-axis) while later SFT model observes higher Pass@1 (x-axis). The interpolated model observe best of both metrics.

In practice, test-time compute is scaled by selecting one of $K$ guesses either by a output reward model (ORM) or Majority Vote. Then we can measure Best@K as

$$\text{Best@K}(x) = \mathbb{E}_{[r_i]_{i=1}^k \sim D(f(x))} \left[ \hat{y}_{i^*} = y \right] \text{ where } i^* = \arg\max_{i \in [K]} \sum_{j=1}^{K} \mathbb{1}\{\hat{y}_i = \hat{y}_j\} \text{ or ORM}(r_i)$$

Notably, Pass@K is equivalent to Best@K using a perfect ORM verifier. As we will observe, WiSE-FT achieves both higher Pass@1 and Pass@K and this directly translates to achieving better Best@K with an ORM verifier and by Majority Vote.

## 3.2 Weight-Space Ensembling (WiSE-FT)

WiSE-FT is a weight-space ensembling technique proposed by Wortsman et al. (2022) to improve the out-of-distribution accuracy of finetuned models at no extra computational cost. In particular, while models tend to achieve better in-distribution performance after finetuning, they tend to be less robust to distribution shift. Surprisingly, by simply interpolating the weights of the finetuned model $w_t$ with the pretrained weights $w_0$

$$w_{\text{WiSE}(t)} = \delta \cdot w_0 + (1 - \delta) \cdot w_t \qquad (2)$$

WiSE-FT can achieve best of both words: the out-of-distribution accuracy of models improves without incurring a drop in in-distribution accuracy. Similar to this philosophy, we apply weight ensembling to achieve both the diverse generation ability of early SFT checkpoints while maintaining the high Pass@1 accuracy of later SFT checkpoints.

## 3.3 Training and Evaluation Pipeline

The majority of our experiments are conducted on Gemma-2-2B and Qwen-2.5-0.5B . We perform SFT on a 30K subset of rephrased augmentations of GSM8k (Cobbe et al., 2021) and MATH (Hendrycks et al., 2021) in MetaMath40k (Yu et al., 2023) for 1710 steps or 10 epochs. We then continue finetuning on another 30K subset of rephrased training questions from MetaMath using Group Relative Policy Optimization (GRPO) with a binary reward of the correctness of the model's final answer. Finally, we evaluate models on GSM8K and MATH500, respectively. To estimate the true Pass@K and Pass@1 marginalized over the distribution of sampled traces, we sample 100 reasoning traces per test example and average over them to estimate Pass@1, i.e. $\rho_x$. Then to calculate Pass@K, we use the theoretical formula $1 - (1 - \rho_x)^K$ in Equation 1. Unless noted otherwise, we employ a naive decoding strategy with top-p threshold 0.9, temperature $T = 0.8$, and top-k with $K = 50$.

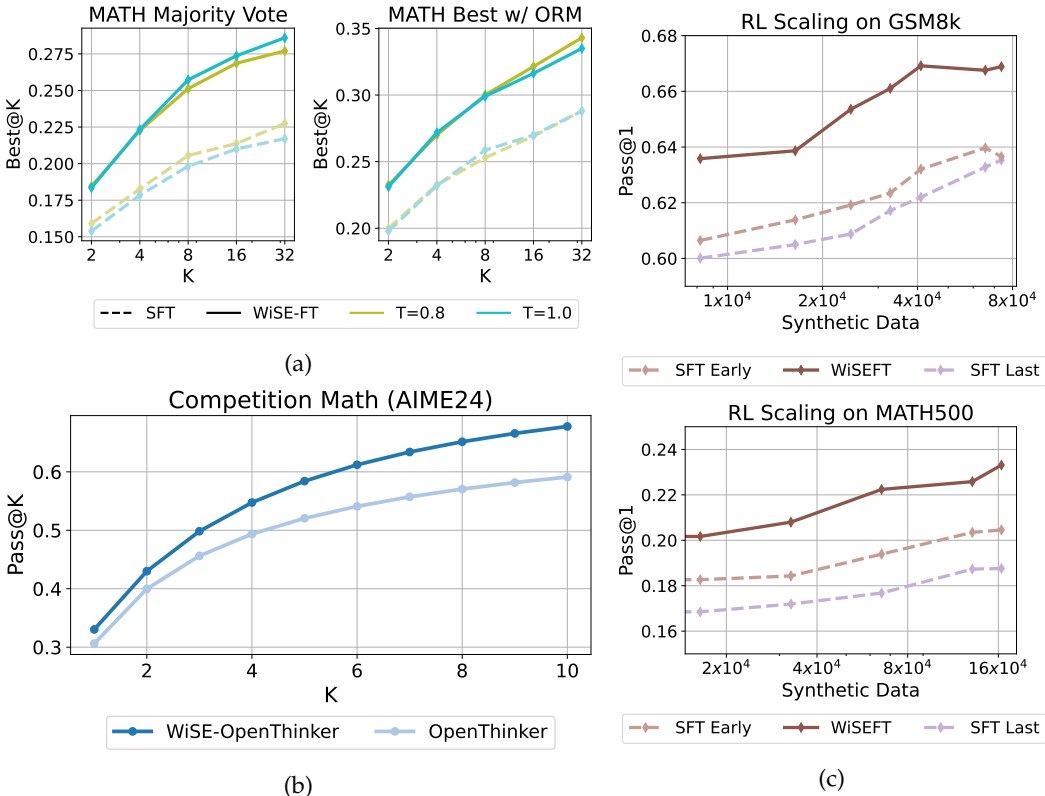

Figure 3: **Downstream Advantages of WiSE-FT: (a) Best@K on MATH500** of the final SFT Gemma-2-2B checkpoint and its WiSE-FT counterpart. **(b) Pass@K on AIME24** WiSE-FT after SFT on general purpose reasoning dataset OpenThoughts-114k achieves higher Pass@K on AIME24. **(c) RL Scaling** Gemma and Qwen SFT checkpoints further tuned by GRPO on GSM8K and MATH, respectively. RL from the final WiSE-FT model achieves higher Pass@1 with less data compared to GRPO starting from both early and late SFT checkpoints.

## 4 Improving Diverse Reasoning Capabilities by WiSE-FT

We first carefully track Pass@K for $K \in \{1, 4, 32\}$ across the SFT trajectory of Qwen-2.5-0.5B and Gemma-2-2B . Similar to findings from Cobbe et al. (2021); Chen et al. (2025), we observe that Pass@1 continues to improve with longer SFT, whereas for larger $K = 4, 32$, Pass@K tends to peak much earlier on in training (in Figure 1, 17, and 19). In other words, while **later** SFT checkpoints achieve higher Pass@1, **earlier** SFT checkpoint achieve higher Pass@K. This tradeoff in model selection is not ideal downstream for test-time scaling.

Building upon this intuition, we propose weight ensembling between earlier and later SFT checkpoints. We apply a variant of WiSE-FT where instead of the pretrained model, we interpolate between the **earliest SFT checkpoint** (in our case, after 1 epoch of training) and the weights of later checkpoint. As shown in Figure 2, we observe a "sweet spot" of interpolation coefficients $\delta \in (0, 1)$ where the WiSE-FT model achieves both higher Pass@1 than the last SFT model and higher Pass@K than the early SFT model. We will fix $\delta = 1/2$, which generally performs decently for all of the datasets we've tested. In fact, after WiSE-FT $w_{\mathsf{WiSE}(t)}$, *both* Pass@1 and Pass@k *grow monotonically with SFT steps t* (see Figure 1).

**Better Test-Time Scaling** This boost in both Pass@1 and Pass@K directly translates to better performance with test-time scaling. We measure Best@K by Majority Vote and by selecting the reasoning trace with highest reward using an off-the-shelf ORM `RLHFlow/Llama3.1-8B-PRM-Deepseek-Data` (Xiong et al., 2024). We evaluate the performance of the last SFT checkpoint with highest Pass@1 versus the corresponding WiSE-FT variant

with $\delta = 1/2$. In Figure 3, we see that the performance gap on MATH500 between the final Gemma-2-2B SFT checkpoint and Wise-FT model widens with larger $K$. The WiSE-FT model achieves $5 - 7\%$ better performance with test-time scaling.

**Better RL Scaling**    WiSE-FT's ability to achieve both high Pass@1 and Pass@K is particularly advantageous for continued RL training where models are further trained by policy gradient methods using self-generated data. In particular, WiSE-FT is able to generate data rich in learning signal (high Pass@1) while still having high coverage over the data space (high Pass@K). We continue training on rephrased training questions of GSM8K and MATH using GRPO paired with a binary reward of the correctness of the final guess. Across runs, we observe that continued RL training starting from the final WiSE-FT model improves performance more stably than finetuning starting from the final SFT checkpoint. Notably the final SFT checkpoint suffers low coverage over the data space, causing Pass@1 to improve slowly. We also try continued RL training from an earlier SFT checkpoint with peak Pass@4 performance. While RL scales better over the early SFT checkpoint in comparison to the final checkpoint, the performance still remains subpar compared to WiSE-FT.

## 4.1   General Purpose Reasoning Models

So far we have studied the effect of WiSE-FT on models tuned on reasoning data for the same specific reasoning task (e.g., train on GSM8k and evaluate on GSM8k). We've additionally tested how well our findings generalize to models trained on general purpose reasoning datasets and tested on a *out-of-distribution* reasoning task. We take Qwen2.5-7B-Instruct and SFT for 5 epochs on OpenThoughts-114k, a high-quality synthetic dataset of math, science, and coding questions paired with DeepSeek-R1 completions, then evaluate its performance on AIME24 competition problems (with ASY code for figures from Muennighoff et al. (2025)). In this setting, the Pass@K trends during SFT on is more subtle. We still observe diversity collapse in Figure 12, but the affect is not strong enough for Pass@K to drop back down. However, we observe that the rate at which Pass@K improves for $K \in \{16, 32\}$ slows down early while Pass@1 grows at a constant rate (Figure 10). We then perform WiSE-FT between the final and earlier checkpoint with higher diversity. We choose early checkpoint at epoch 3 where improvements in Pass@K begin to slow. Similarly, we observe that WiSE-FT improves both Pass@1 and Pass@K in Figure 2.

## 5   Diversity Collapse during Finetuning

In previous sections we alluded to the phenomenon where Pass@K decreases because SFT and RL induces *diversity collapse* in reasoning traces. To verify this hypothesis, we sample 100 traces per test GSM8k problem and measure diversity using three metrics:

1. **Answer Diversity:** The fraction of unique guesses $\hat{y}$ among reasoning traces.
2. **Operation Diversity:** The fraction of unique sequence of arithmetic operations performed among reasoning traces (In GSM8k, each intermediate step consists of a basic arithmetic operation, e.g. $5 + 3 = 8$.).
3. **Semantic Diversity:** The average cosine similarity between the text embeddings of the reasoning traces, computed using Stella-400M-v5 (Zhang et al., 2024a)

As shown in Figure 4, we observe a stark trend where longer SFT on Gemma-2-2B incrementally suffers from clear diversity collapse across all diversity metrics. Specifically, the model places most of its probability mass not only on one particular guess, but on *a single reasoning trace*, as evidenced by the reduced semantic and operation diversity.

## 5.1   Theoretical Discussion of Diversity Collapse During SFT and RL

We assess theoretically why diversity collapse tends to arise during SFT and RL training. Our analysis reveals that while SFT and RL operate on different principles, they share common pathways that lead to reduced generation diversity when optimizing for accuracy.

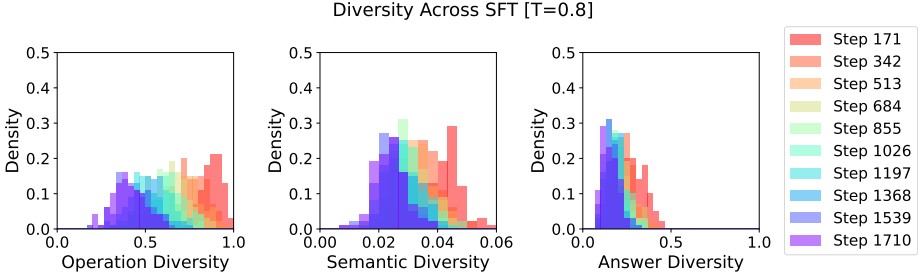

Figure 4: **Diversity Collapse** The answer, semantic, and operation diversity of Gemma-2-2B reasoning traces across GSM8k test examples. Colors map to different SFT checkpoints.

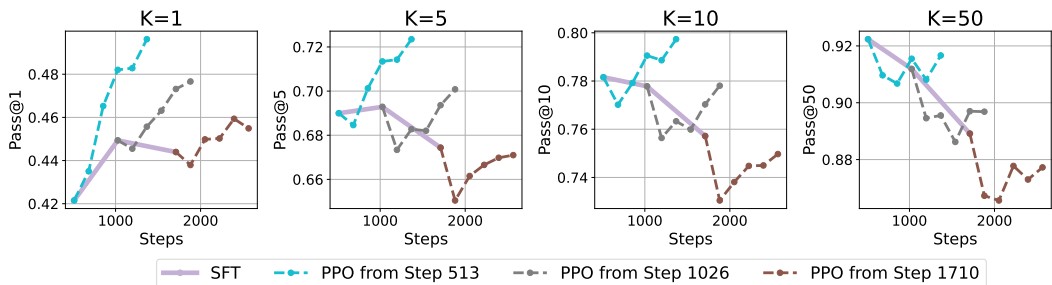

Figure 5: **Pass@k for SFT and RL of Qwen-2.5-0.5B on GSM8K.** The purple solid line measures Pass@K across SFT steps, while the dashed lines correspond to further training different checkpoints by Proximal Policy Optimization (PPO). While Pass@1 continues to improve, Pass@k for larger K can decrease even with RL.

**Diversity Collapse during SFT** Overparameterized models are well-known to exhibit overconfidence in their predictions, an effect that has been studied extensively in classification (Guo et al., 2017). In particular, the model's confidence towards the most likely class $P(\hat{y} = k_{max} \mid x)$ is often much higher than the model's accuracy. In binary classification with linear models $f(x) = \sigma(\langle w, x \rangle)$ and linearly separable training data, gradient descent provably drives the norm of the weights to infinity, causing probabilities to collapse to 0 or 1 (Soudry et al., 2018). We demonstrate this in linear models in Appendix A. A similar phenomenon likely arises in large reasoning models, which may also be prone to overfitting during SFT, ultimately leading to overly confident solutions in spite of limited coverage over the space of traces (Cobbe et al., 2021).

**Diversity Collapse during RL** We further prove why applying reinforcement learning to a low-diversity policy yields suboptimal results—and sometimes even exacerbates diversity collapse—in a discrete bandit setting (see Figure 5). In this scenario, we assume there exist $K$ equally good arms, corresponding to a set of successful strategies, and one bad arm that the policy should learn to avoid. We show two key results in this setting:

1. *Implicit Collapse of Policy Diversity without KL Regularization.* Our analysis demonstrates that when standard reinforcement learning algorithms—REINFORCE and GRPO—are applied without KL regularization, the training dynamics inevitably lead to a collapse in output diversity. Although multiple arms (actions) are equally optimal, the updates become self-enforcing as training progresses. Once one of the good arms is randomly reinforced, its probability increases at the expense of the others, ultimately driving the policy to converge on a single-arm strategy (Theorem C.1).

2. *Diversity Does Not Increase with KL Regularization.* When KL regularization is incorporated to constrain the divergence from the initial policy in REINFORCE, the final policy no longer collapses into a single-arm strategy. However, the diversity of the converged policy **cannot** exceed the initial diversity. Concretely, we show that the probability

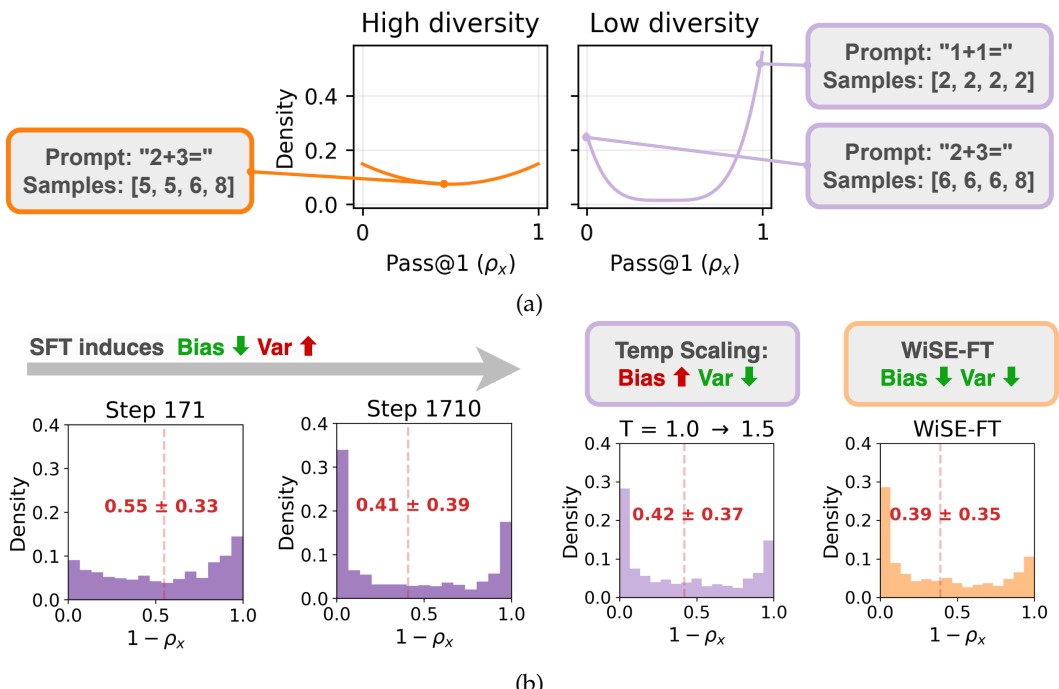

Figure 6: **Histogram of error $1 - \rho_x$** of Gemma-2-2B SFT checkpoints across GSM8k test. SFT progressively decreases bias but increases variance of error i.e., $1 -$ Pass@1, across the test distribution, causing Pass@K to fall. Applying Wise-FT reduces both bias and variance, but temperature scaling trades off decreasing variance with increased bias.

distribution over the good arms remains proportional to the initial distribution when the RL algorithm converges (Theorem C.8). This explains why initializing with a diverse policy is critical for the generalization of reinforcement learning.

# 6 Bias-Variance Tradeoff of Pass@K

So far, we saw a mismatch in growth of Pass@1 and Pass@K during SFT and alluded to the impact of diversity collapse to Pass@K. We now formalize the relationship between Pass@1, Pass@K, and diversity collapse. Notably, we show that the upper bound of *expected* Pass@K over the test distribution can be decomposed into bias and variance quantities.

## 6.1 Diversity Collapse leads to Bimodal Pass@1 Distribution

Consider the expected Pass@K over the entire test distribution $x, y \sim \mathcal{D}$. By Jensen's inequality, we can derive a straightforward upper bound of expected Pass@K that decomposes into the *bias and variance* of $1 - \rho_x$ (See proof in Appendix B). Note that the upper bound falls monotonically with larger bias and variance:

> **Proposition 6.1.** $\mathbb{E}_{x,y\sim\mathcal{D}}\left[\text{Pass@K}(x)\right] \leq 1 - \left((\underbrace{\mathbb{E}_{x,y\sim\mathcal{D}}[1 - \rho_x]}_{\text{Bias}})^2 + \underbrace{\text{Var}(\rho_x)}_{\text{Variance}}\right)^{k/2}$

In Figure 6b, we plot the distribution of error $1 - \rho_x$, estimated using 100 sampled traces, over GSM8K test examples. We notice two trends with longer SFT. First, *bias decreases*, i.e., the expected error shifts towards 0. However, the distribution becomes increasingly bimodal with the densities converging towards the two extremes 0 and 1. As a result, the *variance increases with longer SFT*. This increase in variance directly explains the drop in Pass@k.

The bimodality of the $1 - \rho_x$ distribution means that the Pass@1 of any test problem is either very high or very low. Interestingly, one explanation for the increased bimodality of the distribution of $1 - \rho_x$ is in fact when models suffer from diversity collapse. In other words, a particular guess to be oversampled for each test problem. If the model places high probability on an incorrect guess, Pass@1 is very low. On the other hand, if the model places high probability on the correct guess, Pass@1 is very high. We illustrate this relationship in Figure 6a. All in all, Pass@K can be improved in two ways — either reduce bias by improving Pass@1 *or* reduce variance by increasing diversity.

## 6.2 WiSE-FT vs. Diverse Decoding

While we've proposed inducing diversity by WiSE-FT, another common alternative for inducing diversity is temperature scaling the logits. High temperature smoothens the logits allowing the model to more likely sample low probability tokens. In Figure 1, we see that while high temperatures indeed improve Pass@K, the Pass@K at any SFT timestep notably *never reaches* the Pass@K of our final WiSE-FT model. If temperature scaling also increases diversity, why does WiSE-FT strictly outperform sampling with high temperature? In Figure 6b, we plot the distribution of $1 - \rho_x$ if we sample from the last SFT checkpoint with high temperature $T = 1.5$. As expected, we see that the model reasons more diversely. This smoothens the bimodal peaks and reduces the variance. However, the average accuracy of the model generations also degrades, causing the bias goes back up. We suspect bias-variance tradeoff is inherent in diversity-inducing decoding approaches. For example, min-p (Nguyen et al., 2024) combines temperature scaling with adaptive thresholding to not sample outlier tokens. However, this additional control is unable to *reduce* bias (Figure 16). Surprisingly, WiSE-FT uniquely manages to reduce both bias and variance.

## 7 Discussion

In this work, we investigated the phenomenon of diversity collapse during the training of reasoning models. Our analysis reveals that standard SFT and RL pipelines can deteriorate in Pass@K due to the convergence of model generations toward a single reasoning trace. We demonstrated that WiSE-FT, which interpolates between early and late SFT checkpoints, significantly improves both Pass@1 and Pass@K across multiple math datasets and model scales. This is unlike alternative approaches such as temperature scaling or early stopping, which face an inherent tradeoff. Furthermore, improving on these metrics corresponded with better adaptation to test-time scaling and RL. But other limitations of WiSE-FT may exist at larger scale, which we leave for future work.

Overall, our work reveals the importance of maintaining diversity in reasoning models. Current decoding strategies (e.g., min-p, nucleus, and top-k) are still unable to fully extract a model's capabilities. We estimate that a significant gap, of tens of percent, remains compared to the optimal decoding strategy for Pass@K, i.e., top-K sampling over the model's marginal answer distribution $P(\hat{y} \mid x)$ (see Table 1 and Appendix G). We encourage future works to address downstream limitations more carefully in earlier stages of the training pipeline.

| Method | Pass@2 | Pass@4 |
|---------|--------|--------|
| Nucleus | 0.57 | 0.67 |
| Min-p | 0.57 | 0.67 |
| Top-k | 0.56 | 0.67 |
| Optimal | **0.76** | **0.83** |

Table 1: Best Pass@k of Gemma on GSM8k across SFT checkpoints

## 8 Acknowledgements

We'd like to thank Aviral Kumar, Sean Welleck, Amrith Setlur and Yiding Jiang for insightful discussions about test-time scaling and reinforcement learning. We'd also like to thank Alex Li, Sachin Goyal, and Jacob Springer for their meaningful contribution to our figures and literature review. We gratefully acknowledge support from Apple, Google, Cisco, OpenAI, NSF, Okawa foundation, the AI2050 program at Schmidt Sciences (Grant #G2264481), and Bosch Center for AI.

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

## A SFT in Binary Classification

**Data and Model Setup** We train a linear classifier $f(x) = \langle w, x \rangle$ from random initialization over a binary Gaussian mixture distribution:

$$x \mid y \sim \mathcal{N}(y\mu, I^{d \times d}) \tag{3}$$
$$y \in \{1, -1\} \text{ uniformly} \tag{4}$$

Given a model, we sample predictions, namely $\hat{y} = 1$ with probability $\sigma(\langle w, x \rangle) = (1 + \exp(-\langle w, x \rangle))^{-1}$, or $\hat{y} = 0$. Then, per-example Pass@1 is equal to $\rho_x = \sigma(y \cdot \langle w, x \rangle)$. Similarly, the expected Pass@k is equal to $1 - (1 - \rho_x)^k$.

In our experiment, we train an overparametrized linear classifier over binary Gaussian data mixture $x \mid y \sim \mathcal{N}(y \cdot \frac{1}{\sqrt{d}}\mathbf{1}, \frac{1}{2}I)$ where $y = \{-1, 1\}$ and $d = 1000$. We then evaluate $\rho_x$ of 400 test samples. As training progresses, the distribution of $\rho_x$ over the test data become bimodal due to the norm of $w$ monotonically increasing once it separates the training examples. Similarly, we observe that this leads to a drop in Pass@k while Pass@1 continues to improve.

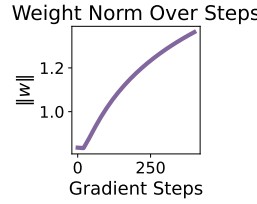

Figure 7: Weight Norm

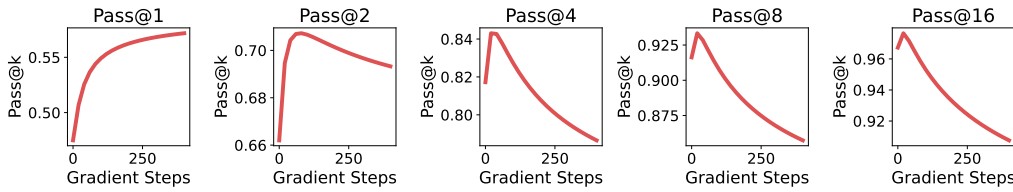

Figure 8: Pass@k across Training in Binary Classification

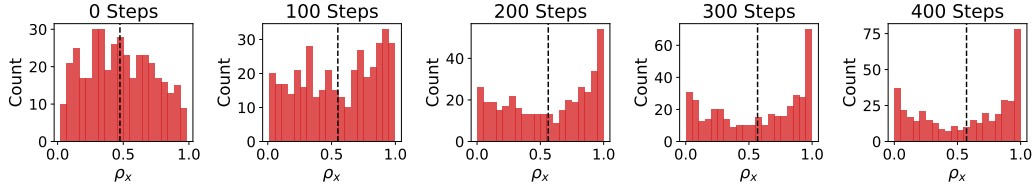

Figure 9: Histogram of $\rho_x$ across training steps

## B Expected Pass@k

**Proposition B.1.**

$$\mathbb{E}_{x,y \sim \mathcal{D}}\left[\text{Pass@K}(x)\right] \leq 1 - ((\mathbb{E}_{x,y \sim \mathcal{D}}[1 - \rho_x])^2 + \text{Var}(\rho_x))^{k/2}$$

*Proof.*

$$\mathbb{E}\left[(1 - \rho_x)^k\right] \geq \mathbb{E}\left[(1 - \rho_X)^2\right]^{k/2} \tag{5}$$

$$= \left(1 - 2\mathbb{E}[\rho_x] + \mathbb{E}\left[\rho_x^2\right]\right)^{k/2} \tag{6}$$

$$= \left(\left(1 - 2\mathbb{E}[\rho_x] + \mathbb{E}[\rho_x]^2\right) + \left(\mathbb{E}\left[\rho_x^2\right] - \mathbb{E}[\rho_x]^2\right)\right)^{k/2} \tag{7}$$

$$= ((1 - \mathbb{E}[\rho_x])^2 + \text{Var}(\rho_x))^{k/2} \tag{8}$$

$\square$

## C   RL Theory

### C.1   Overview

We will prove that in a discrete bandit setting with $K$ equally good arms that is the best arm, both REINFORCE and GRPO without KL regularization will eventually collapse into a single-arm strategy.

We will further prove that, with KL regularization with respect to the initial policy, the converged policy of REINFORCE have the same action distribution as the initial policy when constrained on the set of best arms. Therefore, diversity within good actions will not increase through REINFORCE training.

### C.2   Notations and Setup

Formally we consider the following setting. We consider a $K + 1$-armed bandit, with arms $\{1, 2, \ldots, K + 1\}$. Arms $1, \ldots, K$ are "good," each yielding reward 1, and the other arm is "bad," yielding reward 0. We use a softmax parameterization:

$$p_i \; = \; \frac{e^{\theta_i}}{\sum_{j=1}^{K+1} e^{\theta_j}}, \quad i = 1, \ldots, K + 1.$$

to denote the action distribution. We will use $\theta_i^{(t)}$ to denote the parameter at step $t$.

It is standard to consider using the KL divergence between the current policy with a reference policy (which we set as $p_0$ here) as a regularization term.

$$\mathrm{KL}(p^{(t)}|p^{(0)}) = \sum_{i=1}^{K+1} p_i^{(t)} \log \frac{p_i^{(t)}}{p_i^{(0)}}$$

For REINFORCE, we will consider the following training setup. At step $t$:

1. We sample an arm $I_t$ according to $p(\cdot) = (p_1^{(t)}, \ldots, p_{K+1}^{(t)})$ and receive reward $r_t$

2. We update using policy gradient.

$$\theta_i^{(t+1)} \; = \; \theta_i^{(t)} \; + \; \eta \, r_t \, \nabla_{\theta_i} (\log p_{I_t}^{(t)}) \; - \; \eta \beta \nabla_{\theta_i} \mathrm{KL}(p^{(t)}|p^{(0)}), \quad i = 1, \ldots, K + 1,$$

   where $\eta > 0$ is the step size and $\beta$ is the hyperparameter controlling the strength of KL regularization.

For GRPO, we will consider the following simplified training setup. This is equivalent to the empirical version of GRPO with online sampling.

1. Sample $G$ arms $\{I_t^{(1)}, \ldots, I_t^{(G)}\}$ i.i.d. from the current policy $p(\cdot)$ and receive rewards $r_t^{(g)}$.

2. Compute

$$\mu_t \; = \; \frac{1}{G} \sum_{g=1}^{G} r_t^{(g)}, \quad \sigma_t \; = \; \sqrt{\frac{1}{G} \sum_{g=1}^{G} (r_t^{(g)} - \mu_t)^2},$$

   and define the normalized advantage

$$\tilde{r}_t^{(g)} \; = \; \begin{cases} \dfrac{r_t^{(g)} - \mu_t}{\sigma_t}, & \sigma_t \neq 0, \\ 0, & \sigma_t = 0. \end{cases}$$

   We will skip the update if $\sigma_t = 0$.

3. Update each parameter $\theta_i$ via

$$\theta_i \; \leftarrow \; \theta_i \; + \; \frac{\eta}{G} \sum_{g=1}^{G} \tilde{r}_t^{(g)} \nabla_{\theta_i} (\log p_{I_t^{(g)}}^{(t)}) \; - \; \eta \beta \nabla_{\theta_i} \mathrm{KL}(p^{(t)}|p^{(0)}). \quad i = 1, \ldots, K + 1,$$

**C.3 Implicit Diversity Collapse without KL regularization**

**Theorem C.1** (Collapse to Deterministic Policy). *Under REINFORCE or GRPO updates without KL regularization ($\beta_0 = 0$), given a sufficient small $\eta$, with probability 1:*

$$\limsup_{t \to \infty} \max_{i \in [K]} p_i^{(t)} = 1.$$

*Thus, the policy collapses to a single-arm strategy during training.*

*Proof.* The proof is two-fold.

Using Lemma C.3 and C.4, we can show that bad arm probability diminishes,

$$\lim_{t \to \infty} p_{K+1}^{(t)} = 0$$

We will then define a property named Self-enforcing Stochastic

**Definition C.2** (Self-enforcing Stochastic Policy Update Rule). *We define three properties of policy update rule that will lead to diversity collapse*

1. *The policy update takes the form of $\sum_{k=1}^{B} A_k \nabla \log p_i(\theta_{i_k})$ where $i_k$ is the k-th sampled arm in the batch and $A_k$ is a function determined by (i) the sum of reward $\sum_{i=1}^{K} r_{i_k}$ with in the batch ; (ii) the reward $r_{i_k}$ and (iii) the batch size B.*

2. *A policy update rule is said to be self-enforcing, if $\mathbb{E}[\theta_i^{(t+1)} - \theta_i^{(t)}]$ is monotonous with $\theta_i^{(t)}$ for all $i \in [K]$ and t. Further $\mathbb{E}[\theta_i^{(t+1)} - \theta_i^{(t)}]$ is non-positive if $i \geq K + 1$ and is non-negative if $i \leq K$.*

3. *A policy update rule is said to be self-enforcing stochastic if it is self-enforcing and there exists constants $C_1, C_2 > 0$ such that for any $\epsilon > 0$, whenever the current policy satisfies $\max_{i \in [K]} p_i^{(t)} \in [1/2K, 1 - \epsilon]$ (i.e., no single good arm dominates), for $i^* = \arg\max_{i \in [K]} p_i^{(t)}$ the conditional second moment of the parameter updates for every arm $i \in [K + 1]$ and $i \neq i^*$ satisfies:*

$$\mathbb{E}\left[ \left( \left( \theta_i^{(t+1)} - \theta_i^{(t)} \right) - \left( \theta_{i^*}^{(t+1)} - \theta_{i^*}^{(t)} \right) \right)^2 \mid \theta^{(t)} \right] \geq C_1 \epsilon^2.$$

*and*

$$|\theta_i^{(t+1)} - \theta_i^{(t)}| < C_2$$

Lemma C.5 shows that for any self-enforcing stochastic policy update rule, the final policy collapses into a single-arm policy.

Using Lemma C.6 and C.7, we can show that REINFORCE and GRPO are self-enforcing stochastic policy update rules when bad arm probability is lower than $1/2$. The proof is then complete. $\qquad\square$

**Lemma C.3** (Bad Arm Probability Diminishes Using REINFORCE). *Under the REINFORCE algorithm without KL regularization ($\beta = 0$), $\lim_{t \to \infty} p_{K+1}^{(t)} = 0$ almost surely.*

*Proof.* We can first simplify the REINFORCE update rule to

$$\theta_i^{(t+1)} = \theta_i^{(t)} + \eta r_t (\mathbf{1}(I_t = i) - p_i^{(t)}), \quad i = 1, ..., K + 1.$$

Noted that $\sum_i \theta_i^{(t)}$ will not change with t, WLOG, assume

$$\sum_i \theta_i^{(t)} = 0.$$

Because $r_{K+1} = 0$, we can then assume without loss of generality, for all $t$, $I_t \leq K$.

This then suggests that

$$\theta_{K+1}^{(t+1)} = \theta_{K+1}^{(t)} - \eta p_{K+1}^{(t)}$$

monotonically decrease.

For any $\epsilon$, if $p_{K+1}^{(t)} > \epsilon$ holds for infinite $t$, then there exists $t_0$, where $\theta_{K+1}^t < \log \epsilon$ for any $t > t_0$. For any $t > t_0$, there exists $i \in [K]$, such that $\theta_i^{(t)} > 0$. This then suggests that

$$p_{K+1}^{(t)} \leq \exp(\theta_{K+1}^{(t)} - \theta_i^{(t)}) \leq \epsilon.$$

This leads to a contradiction. The proof is then complete. □

**Lemma C.4** (Bad Arm Probability Diminishes Using GRPO). *Under the GRPO algorithm without KL regularization ($\beta = 0$),$\lim_{t \to \infty} p_{K+1}^{(t)} = 0$ almost surely.*

*Proof.* For GRPO, we can show that $\tilde{r}_t^{(g)}$ is negative iff $I_t^{(g)} = K + 1$. Therefore, we can show that $\theta_{K+1}^{(t)}$ monotonically decreases, similar to the case in REINFORCE.

If $p_{K+1}^{(t)} > \epsilon$ holds for some $t$, one can prove that $\theta_{K+1}^{(t)}$ will decrease by a constant depending on $\epsilon$ in expectation. Therefore, following the same line as in C.3, we can prove that $\lim_{t \to \infty} p_{K+1}^{(t)} = 0$ almost surely. □

**Lemma C.5** (Collapse Happens for All Self-enforcing Stochastic Policy Update Rule). *Consider a policy update process that is self-enforcing stochastic (Definition C.2), then $\limsup_{t \to \infty} \max_{i \in [K]} p_i^{(t)} = 1$ almost surely.*

*Proof.* We will inductively prove that for different $K$ the following induction hypotheses, for any $\epsilon, \delta > 0$, there exists $T_{\epsilon, \delta, K} > 0$,

$$\Pr(\max_{t < T_{\epsilon, \delta, K}} \max_{i \in [K]} p_i^{(t)} < 1 - \epsilon) < \delta.$$

We first consider the case where $K = 2$.

Consider the stopping time,

$$\tau_\epsilon = \arg\min_t \max_{i \in [K]} p_i^{(t)} > 1 - \epsilon$$

For any $\mathcal{I} = \{1, 2\}$,define $\Delta_{\mathcal{I}}^t = \max_{j \in [K]} \theta_j^t - \min_{j \in \mathcal{I}} \theta_i^t$.

Assume $\theta_{i*}^t = \max_{j \in [K]} \theta_j^t$, because $|\mathcal{I}| \geq 2$, there exists $i \neq i^*$, $\min_{j \in \mathcal{I}} \theta_i^t > 0$. We will show three properties of $\Delta_{\mathcal{I}}^t$

First $\Delta_{\mathcal{I}}^{(t)}$ is a submartingale defined on the filtration of the distribution of $\theta^{(t)}$ because

$$\mathbb{E}[\Delta_{\mathcal{I}}^{(t)} \mid \theta_t] - \Delta_{\mathcal{I}}^{(t-1)} > \mathbb{E}[(\theta_{i*}^{t+1} - \theta_{i*}^t) - (\theta_i^{t+1} - \theta_i^t) \mid \theta_t] > 0.$$

as the policy is self-enforcing.

Further $\Delta_{\mathcal{I}}^{(t)}$ has bounded growth of $2C_2$ as

$$|\max_{j \in [K]} \theta_j^{t+1} - \max_{j \in [K]} \theta_j^t| < C_2.$$

$$|\min_{j \in \mathcal{I}} \theta_j^{t+1} - \max_{j \in \mathcal{I}} \theta_j^t| < C_2.$$

Furthermore, the second-momentum of $\Delta_{\mathcal{I}}^{(t)}$ needs to increase with $t$ by a constant for any $t < \tau_\epsilon$.

$$\mathbb{E}[(\Delta_{\mathcal{I}}^{(t+1)})^2 \mid \theta_t] \geq (\Delta_{\mathcal{I}}^{(t)})^2 + \mathbb{E}[(\Delta_{\mathcal{I}}^{(t+1)} - \Delta_{\mathcal{I}}^{(t)}))^2 \mid \theta_t]$$
$$\geq (\Delta_{\mathcal{I}}^{(t)})^2 + C_1\epsilon^2.$$

When $t < \tau_\epsilon$, it holds that $\Delta_{\mathcal{I}}^{(t)} < \log \frac{2}{\epsilon}$, otherwise we can prove that

$$\max_{i,j \in \{1,2\}} p_i/p_j = \exp(\Delta_{\mathcal{I}}^{(t)}) > \frac{2 - 2\epsilon}{\epsilon}. \implies \max_{i \in \{1,2\}} p_i > 1 - \epsilon.$$

This is a contradiction. Further, by Martingale inequality, we have that

$$\mathbb{E}[\left(\Delta^{\min\{t,\tau_\epsilon\}}\right)^2] > \mathbb{E}[\left(\Delta^0\right)^2] + C_1\epsilon^2 \mathbb{E}[\min\{t,\tau_\epsilon\}]$$

Further, as $\Delta^t$ has bounded growth, we have that

$$\mathbb{E}[\left(\Delta^{\min\{t,\tau_\epsilon\}}\right)^2] < (\log \frac{2}{\epsilon} + 2C_2)^2.$$

This implies $\mathbb{E}[\min\{t, \tau_\epsilon\}] < \frac{(\log \frac{2}{\epsilon} + 2C_2)^2}{C_1\epsilon^2}$ for all $t$, this implies

$$\mathbb{E}[\tau_\epsilon] < \frac{(\log \frac{2}{\epsilon} + 2C_2)^2}{C_1\epsilon^2}.$$

Further, by Markov inequality, if we choose

$$T_{\epsilon,\delta,2} = \frac{(\log \frac{2}{\epsilon} + 2C_2)^2}{C_1\epsilon^2\delta}.$$

then,

$$\Pr(\tau_\epsilon > T_{\epsilon,\delta,2}) < \frac{\mathbb{E}[\tau_\epsilon]}{T_{\epsilon,\delta,2}} < \delta.$$

This concludes the proof for $K = 2$.

Now assuming the result holds for $K - 1$ and consider the case for $K$, First, we choose a small enough constant $C_{\delta,\epsilon,K,N} > 0$, such that when $p_{K-1}^{(0)} < C_{\delta,\epsilon,K,N}$, the following two random processes are close:

- Running the algorithm for $N$ steps on the $K$ arms bandit yields $\theta_i^{(t)}, i \in [K]$

- Running the algorithm for $N$ steps on a $K - 1$ arms bandit yields $\tilde{\theta}_i^{(t)}, i \in [K - 1]$ with $\tilde{\theta}_i^{(0)} = \theta_i^{(0)}, i < K - 1$ and $\tilde{\theta}_{K-1}^{(0)} = \theta_K(0)$

and there exists a joint measure on $\theta$ and $\tilde{\theta}$ such that

$$\forall i \in [K-2], t < N, \Pr(|p_i^t - \tilde{p}_i^t| > \epsilon/2) < \delta/6.$$
$$\Pr(|p_K^t - \tilde{p}_{K-1}^t| > \epsilon/2) < \delta/6.$$
$$\Pr(|p_K^t - p_K^0| > \epsilon/2) < \delta/6.$$

This joint measure is constructed by choosing the corresponding arm for two process at each sampling step as long as the sampled arm is not $K$ and uses the uniform convergence on $\nabla \log_\theta p_i$. Now following the same argument at $K = 2$, we can show that there exists $\tilde{T}_{\epsilon,\delta,K}$ such that

$$\Pr(\exists t < \tilde{T}_{\epsilon,\delta,K}, \min_{t \in [K]} p_t < C_{\delta,\epsilon,K,T_{\epsilon/2,\delta/2,K-1}}) > 1 - \delta/2.$$

Then we can invoke the induction hypothesis and uses the coupling shown above to show that if we choose $T_{\epsilon,\delta,K} = \tilde{T}_{\epsilon,\delta,K} + T_{\epsilon/2,\delta/2,K-1}$, then there exists a time step that one arm has probability higher than $1 - \epsilon$ with probability at least $1 - \delta$.

$\square$

**Lemma C.6.** *The REINFORCE algorithm without KL regularization ($\beta = 0$) is self-enforcing stochastic (Definition C.2) once $p_{K+1}^{(t)} < 1/2$.*

*Proof.* The REINFORCE algorithm is self-enforcing because

$$\mathbb{E}[\theta_i^{(t+1)} - \theta_i^{(t)}] = \eta p_i (r_i - \sum_{j \in [K+1]} p_j r_j).$$

Further,

$$|\theta_i^{(t+1)} - \theta_i^{(t)}| \leq 1$$

and if we consider the distribution of $\Delta_{i,i^*,t} = \frac{\left(\theta_i^{(t+1)} - \theta_i^{(t)}\right) - \left(\theta_{i^*}^{(t+1)} - \theta_{i^*}^{(t)}\right)}{\eta}$, it holds that

$$\Delta_{i,i^*,t} = r_{I_t} \left(\mathbf{1}(i = I_t) - \mathbf{1}(i^* = I_t) - p_i + p_{i^*}\right)$$

$$\Pr\left(\Delta_{i,i^*,t} = -1 - p_i + p_i^*\right) \geq \Pr\left(I_t = i^*\right) = p_{i^*}$$

Therefore

$$\mathbb{E}[\Delta_{i,i^*,t}^2] \geq p_{i^*}(-1 - p_i + p_i^*)^2$$
$$\geq p_{i^*}(1 - p_{i^*})^2 \geq \frac{\epsilon^2}{2K}.$$

This then concludes the proof with $C_1 = \eta/2K$ and $C_2 = \eta$. $\square$

**Lemma C.7.** *The GRPO algorithm without KL regularization ($\beta = 0$) is self-enforcing stochastic (Definition C.2) once $p_{K+1}^{(t)} < 1/2$.*

*Proof.* The GRPO algorithm is self-enforcing because

$$\mathbb{E}[\theta_i^{(t+1)} - \theta_i^{(t)}] = \eta \mathbb{E}[\tilde{r}_t^{(g)} \left(\mathbf{1}(I_t^{(g)} = i) - p_i^{(t)}\right)] = \eta \mathbb{E}[\tilde{r}_t^{(g)} \mathbf{1}(I_t^{(g)} = i)] = \eta \mathbb{E}_{\mu_t}[\mathbb{E}[\tilde{r}_t^{(g)} \mathbf{1}(I_t^{(g)} = i)|\mu_t]].$$

Noted that $\mathbb{E}[\tilde{r}_t^{(g)} \mathbf{1}(I_t^{(g)} = i)|\mu_t]$ is monotonous with $p_i$, hence monotonous with $\theta_i$.

Further

$$|\theta_i^{(t+1)} - \theta_i^{(t)}| \leq \eta \max_g |\tilde{r}_t^{(g)} \left(\mathbf{1}(I_t^{(g)} = i) - p_i^{(t)}\right)|$$

$$\leq \eta \max_g |\tilde{r}_t^{(g)}| \leq \eta\sqrt{G}.$$

Now we only need to lower bound the second momentum of

$$\Delta_{i,i^*,t} = \frac{\left(\theta_i^{(t+1)} - \theta_i^{(t)}\right) - \left(\theta_{i^*}^{(t+1)} - \theta_{i^*}^{(t)}\right)}{\eta}$$

.

Noted that

$$\theta_i^{(t+1)} - \theta_i^{(t)} = \frac{\eta}{G} \sum_{g=1}^{G} \tilde{r}_t^{(g)}\mathbf{1}(I_t^{(g)} = i).$$

It holds that

$$\sigma_t = \sqrt{\frac{1}{G}\sum_g (r_t^g - \mu)^2} = \sqrt{\frac{1}{G}\sum_g r_t^g - 2\mu r_t^g + \mu^2} = \sqrt{\mu - \mu^2}.$$

Therefore when $r_t^{(g)} > 0$,

$$\tilde{r}_t^{(g)} = \frac{r_t^{(g)} - \mu_t}{\sigma_t} = \frac{1 - \mu_t}{\sigma_t} = \sqrt{\frac{1 - \mu_t}{\mu_t}} \geq \sqrt{\frac{1}{G-1}}.$$

Because all $\tilde{r}_t^{(g)}$ are the same when $r_t^{(g)} > 0$, it holds that when $i \in [K]$,

$$\Delta_{i,i^*,t}^2 = \frac{1}{G}\frac{1 - \mu_t}{\mu_t}\left(\sum_{g=1}^{G}\mathbf{1}(I_t^{(g)} = i) - \mathbf{1}(I_t^{(g)} = i^*)\right)^2$$

$$\geq \frac{1}{G(G-1)}\left(\sum_{g=1}^{G}\mathbf{1}(I_t^{(g)} = i) - \mathbf{1}(I_t^{(g)} = i^*)\right)^2.$$

This then implies

$$\mathbb{E}[\Delta_{i,i^*,t}^2] \geq \frac{1}{G(G-1)}\mathbb{E}\left[\left(\sum_{g=1}^{G}\mathbf{1}(I_t^{(g)} = i) - \mathbf{1}(I_t^{(g)} = i^*)\right)^2 \Bigg| \mu_t \neq 1,0\right]$$

One can without loss of generality assume $I_t^{(G)} = K + 1$ and show that

$$\mathbb{E}[\Delta_{i,i^*,t}^2] \geq \frac{1}{G(G-1)}\mathbb{E}\left[\left(\sum_{g=1}^{G-1}\mathbf{1}(I_t^{(g)} = i) - \mathbf{1}(I_t^{(g)} = i^*)\right)^2\right]$$

$$\geq \frac{1}{G}\mathbb{E}\left[\left(\mathbf{1}(I_t^{(1)} = i) - \mathbf{1}(I_t^{(1)} = i^*)\right)^2\right] = \frac{p_i + p_i^*}{G} \geq \frac{1}{2KG}.$$

When $i \neq K$, noted that $\left(\theta_i^{(t+1)} - \theta_i^{(t)}\right) - \left(\theta_{i^*}^{(t+1)} - \theta_{i^*}^{(t)}\right) > \left(\theta_i^{(t+1)} - \theta_i^{(t)}\right) > 0$. Therefore, a similar bound can show that $\mathbb{E}[\Delta_{i,i^*,t}^2] > \frac{1}{2KG}$. This then concludes the proof with $C_1 = \eta/2KG$ and $C_2 = \sqrt{G}$.

$\square$

## C.4 Diversity Never Improves with KL regularization

**Theorem C.8** (Diversity Preservation under KL Regularization). *With $p_0$ as the initial policy and KL-regularization hyperparameter $\beta > 0$, if the REINFORCE process converges to policy $p^*$. Then, $p^*$ satisfies:*

$$\frac{p^*(i)}{\sum_{j=1}^{K} p^*(j)} = \frac{p_0(i)}{\sum_{j=1}^{K} p_0(j)} \quad \forall i \in \{1, \ldots, K\}.$$

*Consequently, the distribution over the optimal arms under $p^*$ matches the initial distribution $p_0$ restricted to these arms and renormalized.*

*Proof.* Using policy gradient theorem, we know that the converged policy $p^*$ and corresponding parameter $\theta^*$ satisfy that,

$$\nabla_\theta \left[ \sum_{i=1}^{K+1} r_i p_i + \beta \mathrm{KL}\left(p|p^0\right) \right] \Bigg|_{\theta=\theta^*} = 0$$

This then suggests that for any $k$

$$r_k p_k^* - p_k^* \sum_{i=1}^{K+1} r_i^* p_i^* + \beta \sum_{i=1}^{K+1} \nabla_{\theta_k}[p_i \log p_i - p_i \log p_i^0] = 0$$

This is equivalent to

$$r_k p_k^* - p_k^* \sum_{i=1}^{K+1} r_i^* p_i^* + \beta \sum_{i=1}^{K+1} (\mathbf{1}(i=k) - p_k^*)p_i^*(\log p_i^* + 1 - \log p_i^0) = 0$$

Simplifying

$$r_k + \beta(\log p_k^* + 1 - \log p_0) = \sum_{i=1}^{K+1} r_i^* p_i^* + \beta \sum_{i=1}^{K+1} p_i^*(\log p_i^* + 1 - \log p_i^0)$$

For all $k \in [K]$, we know that $r_k$ is equivalent, therefore, $\frac{p_k^*(i)}{p_0^*(i)}$ is a constant for $k \in [K]$, concluding our proof. $\qquad \square$

## C.5 Technical Lemma

**Lemma C.9.** *For $x \in \mathbb{R}, |x| < C$, it holds that*

$$\exp(x) > 1 + x + A_C x^2$$

*here $A_C = \frac{\exp(-C)+C-1}{C^2}$.*

*Proof.* Define $g(x) = \frac{\exp(x)-1-x}{x^2}$, this function monotonically increase when $x < 0$. $\qquad \square$

# D  Open-Thoughts Evaluation

We full finetune Qwen2.5-7B-Instruct over OpenThoughts-114k for 5 epochs using BF16 and AdamW with hyperparameters `learning-rate=1e-5, batch-size=128, warmup=150 steps`. We sample 40 reasoning traces with temperature set to 0.7 for each of the 30 problems in AIME24. Then we evaluate the following quantities.

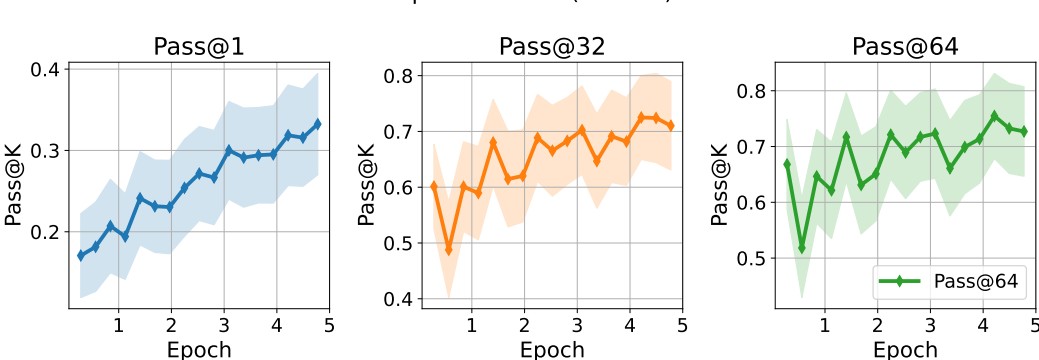

Figure 10: Pass@K Evaluated on AIME24 over OpenThoughts-114K SFT checkpoints. We plot the expected Pass@K $\pm$ SD. Note that the rate at which Pass@K slows down while Pass@1 improves at a constant rate. Furthermore, note that the confidence interval of Pass@1 widens, meaning the variance increases during SFT.

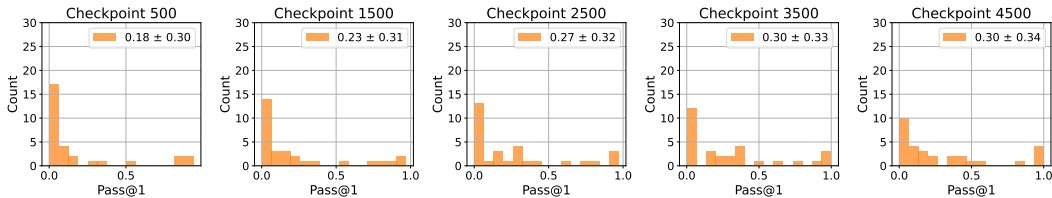

Figure 11: Histogram of Pass@1 over AIME24. Variance of Pass@1 increases over finetuning on OpenThoughts-114K. We note that since AIME24 only has 30 questions, the density plot may not be completely reliable.

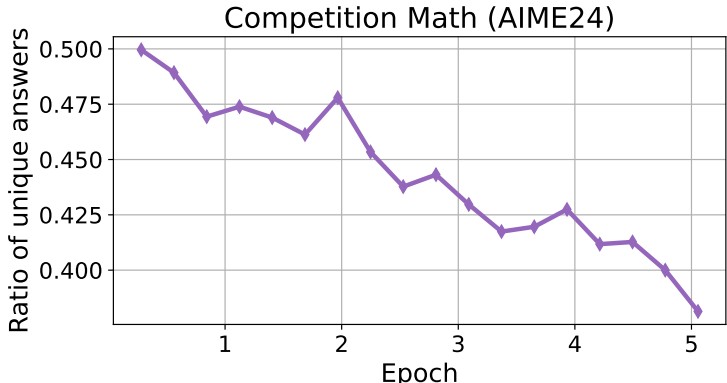

Figure 12: We plot the average number of unique answers sampled over the total number samples i.e. $\frac{\left|\{y_i\}_{i=1}^n\right|}{n}$. Model samples less diverse number of answers as SFT progresses.

# E   Interpolation Coefficients

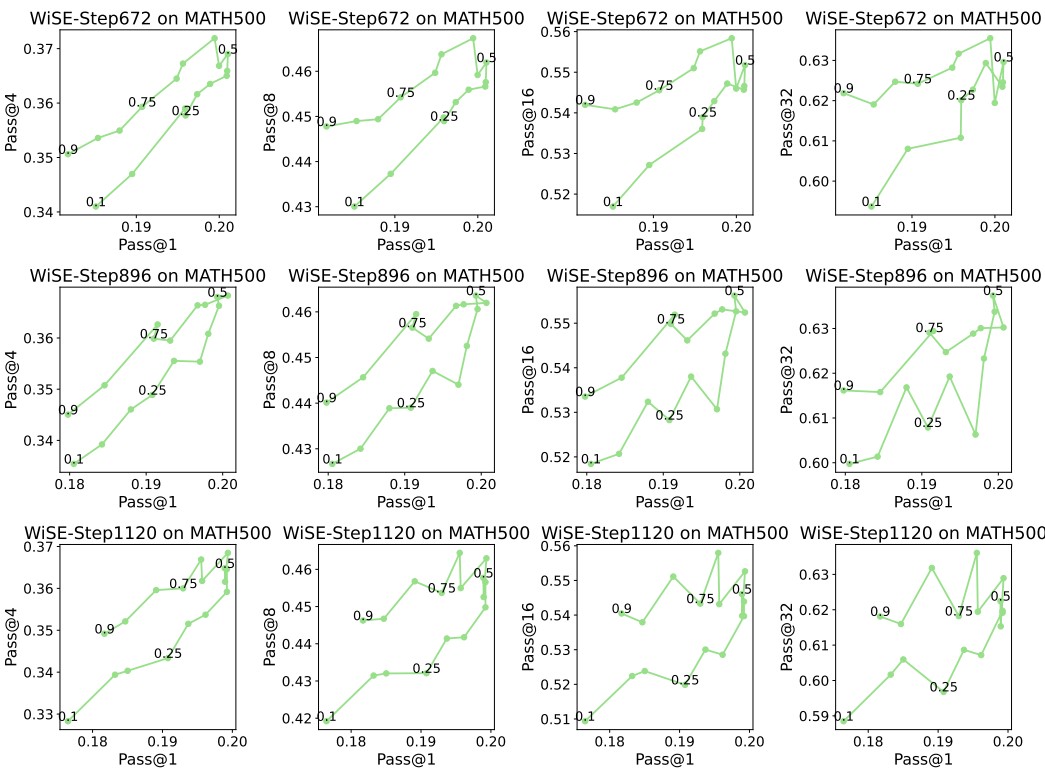

Figure 13: Pass@1 versus Pass@K of WiSEFT of Qwen-2.5-0.5B trained and evaluated on MATH500. We interpolate between model $w_0$ at Step 112 with $w_t$ for $t \in [672, 896, 1120]$ as $\delta w_0 + (1 - \delta) w_t$ where $\delta \in [0.1, 0.9]$.

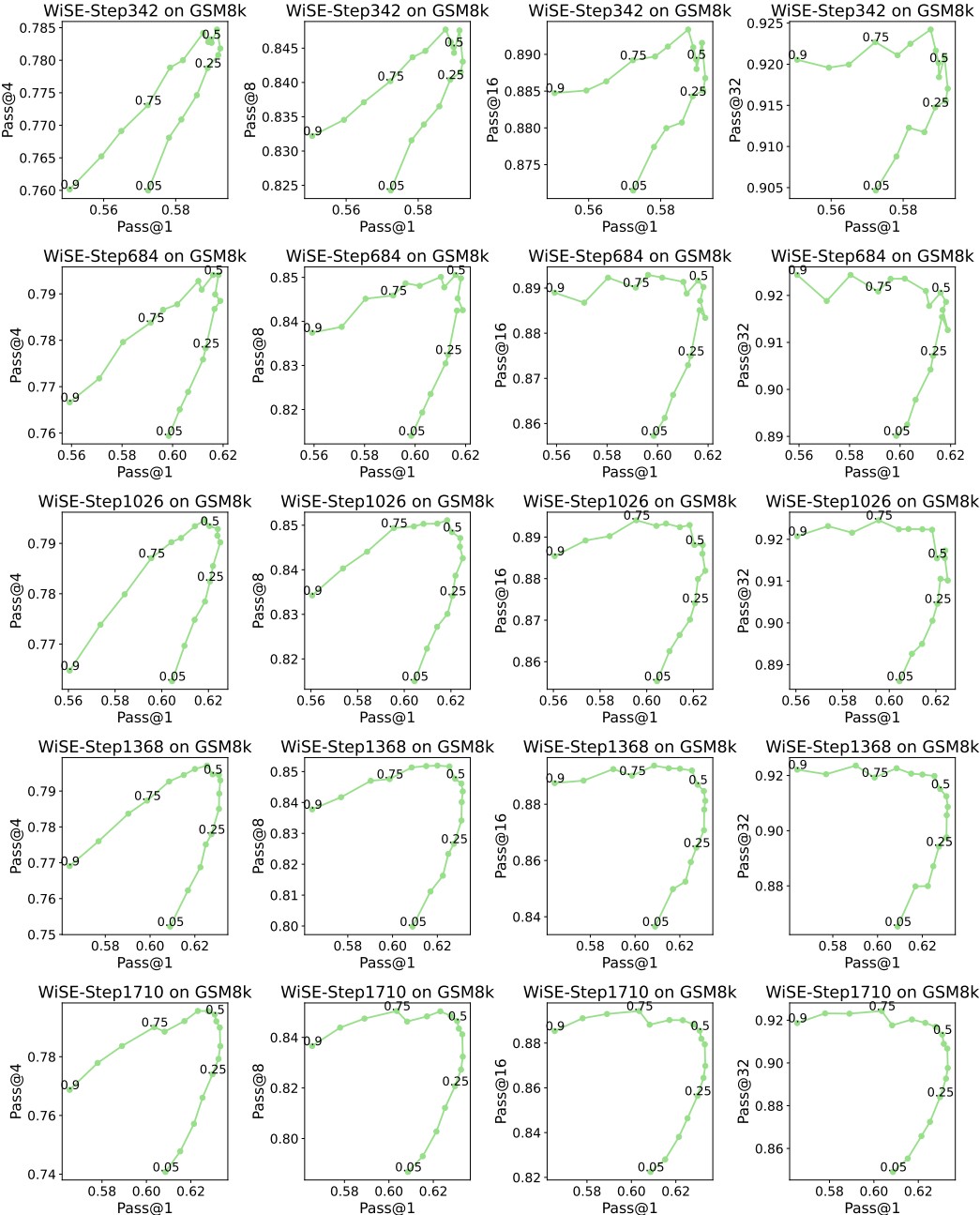

Figure 14: Pass@1 versus Pass@K of WiSEFT of Gemma-2-2B trained and evaluated on GSM8K. We interpolate between model $w_0$ at Step 171 with $w_t$ for $t \in [342, 684, 1026, 1368, 1710]$ as $\delta w_0 + (1 - \delta) w_t$ where $\delta \in [0.05, 0.9]$.

# F   Measuring Diversity of Traces

We measure the *diversity* of the 100 sampled traces of Gemma-2-2B across GSM8k test. We measure diversity in terms of 3 different measures.

**Output Diversity** The cardinality or number of unique answers in the set of all model outputs $|\{\hat{y}_1, \hat{y}_2, \ldots, \hat{y}_n\}|$ over the total number of traces.

**Operation Diversity** In GSM8k, each intermediate step consists of basic arithmetic operations, e.g. $5 + 3 = 8$. We may simply map each of the traces to the sequence of arithmetic operations the model steps through, i.e. $r_i \rightarrow [o_1, o_2, \ldots, o_t]$. This mapping is extracted by code. Then, given this set, we measure unique sequence of operations over the number of total traces.

**Semantic Diversity** We measure the similarity of trace using cosine similarities between the text-embeddings (Bilmes, 2022; Yu et al., 2023).

## F.1   Does temperature increase diversity?

Temperature does increase diversity, but it also increases the chances of sampling outlier answers.

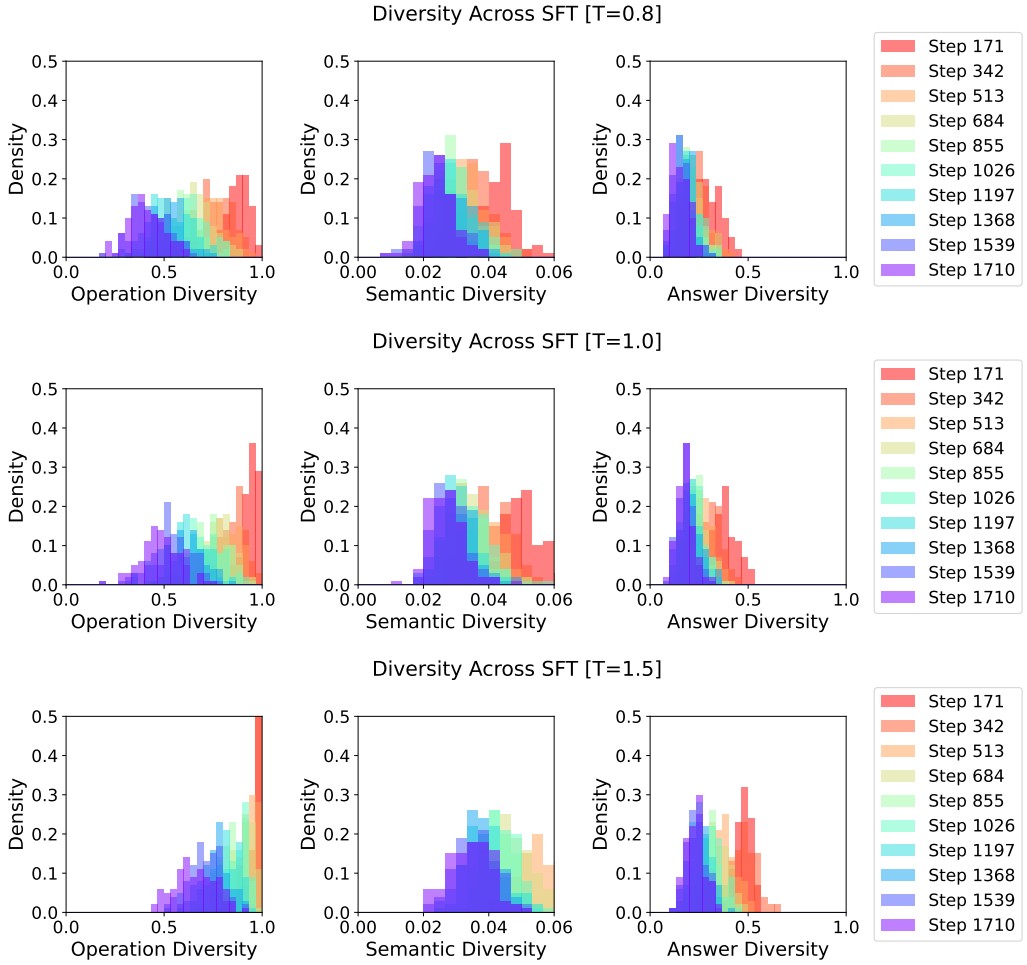

Figure 15: Diversity of traces sampled with Temperature $\in \{0.8, 1.0, 1.5\}$ for Gemma-2-2B SFT checkpoints on GSM8k

### F.2 How well do token-level diverse decoding strategies compare with optimal strategy with oracle?

**Hyperparameter Tuning Details** We grid search for optimal temperature for all baselines over $T = [0.8, 1.0, 1.2, 1.5, 1.8]$. For nucleus, we choose the best cutoff threshold between $[0.8, 0.9, 0.95]$. For min-p, we choose the best probability threshold between $[0.01, 0.05, 0.1]$. For tokenwise top-k, we choose best k between $[12, 25, 50]$.

| Decoding Strategy | Pass@2 | Pass@4 | Pass@8 |
|:---:|:---:|:---:|:---:|
| Naive | 0.565 | 0.666 | 0.760 |
| Nucleus | 0.566 | 0.668 | 0.757 |
| Min-p | 0.566 | 0.668 | 0.760 |
| Top-k | 0.563 | 0.666 | 0.756 |
| Top-k w/Oracle | 0.760 | 0.832 | 0.901 |

Table 2: Best Pass@k of Sampling Strategies for Qwen-2.5-0.5B over SFT checkpoints

| Decoding Strategy | Pass@2 | Pass@4 | Pass@8 |
|:---:|:---:|:---:|:---:|
| Naive | 0.547 | 0.648 | 0.737 |
| Nucleus | 0.528 | 0.617 | 0.694 |
| Min-p | 0.550 | 0.655 | 0.744 |
| Top-k | 0.538 | 0.646 | 0.738 |
| Top-k w/Oracle | 0.730 | 0.814 | 0.878 |

Table 3: Pass@k of Sampling Strategies for Qwen-2.5-0.5B at Last SFT Checkpoint

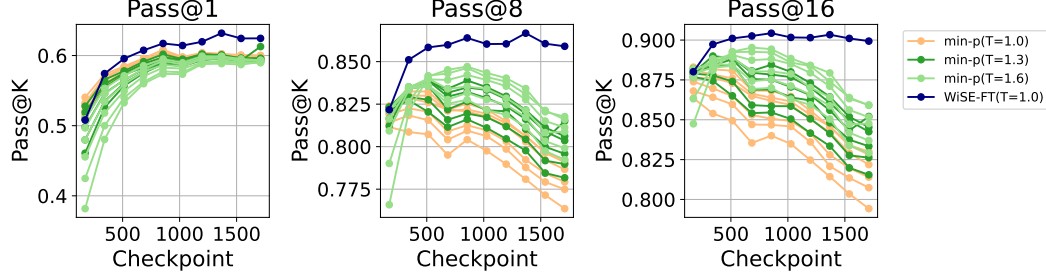

Figure 16: Pass@K over different Min-P thresholds $\gamma \in [0, 0.3]$ and temperatures $T \in [1, 1.6]$ for Gemma-2-2B finetuned on GSM8K. Generally, no min-p threshold paired with high temperature T = 1.6 (in light green) is able to surpass the Pass@1 of T = 1 with best min-p threshold (in orange). In other words, unlike WiSE-FT which increases both Pass@1 and Pass@K, Pass@1 tends to still decrease for the diverse decoding strategy of applying min-p with high temperature.

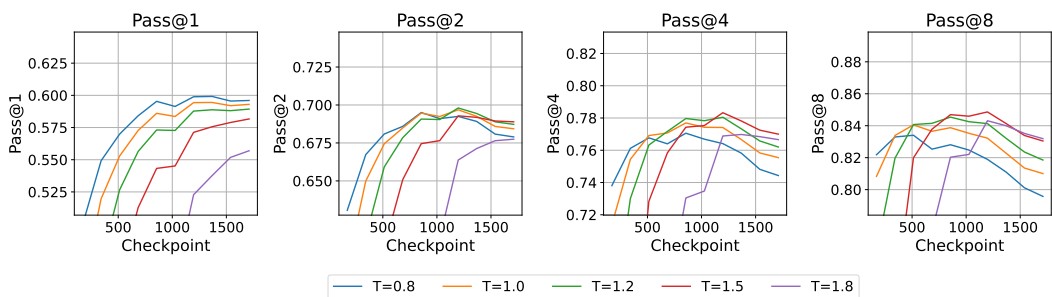

Figure 17: Pass@k of Gemma-2-2B GSM8k Naive Sampling with Replacement

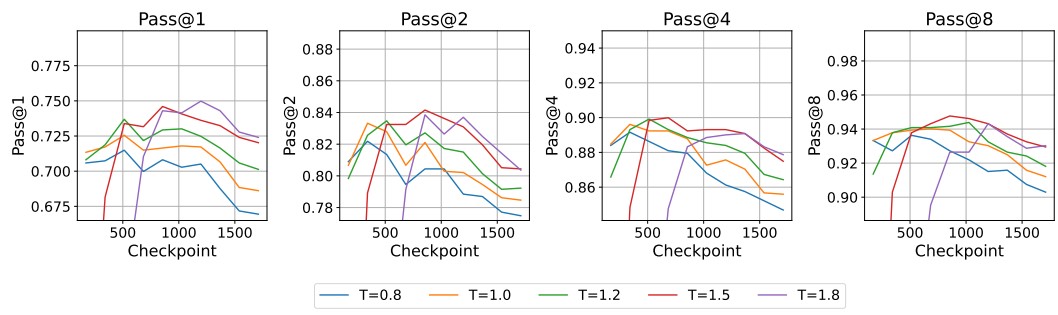

Figure 18: Pass@k of Gemma-2-2B GSM8k Oracle Top K Sampling

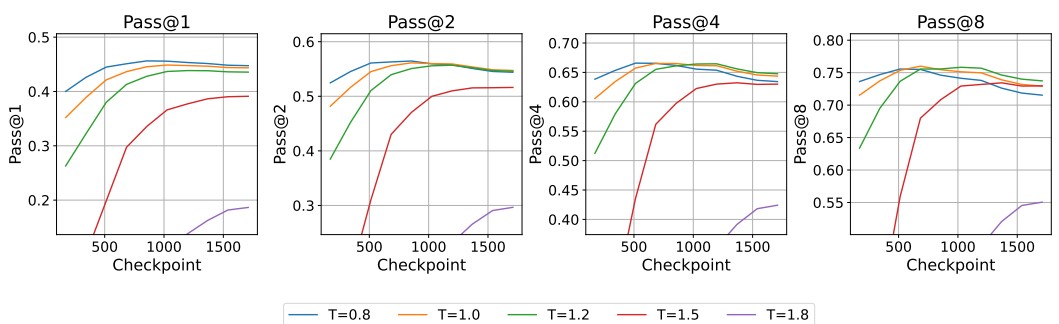

Figure 19: Pass@k of Qwen-2.5-0.5B GSM8k Naive Sampling with Replacement

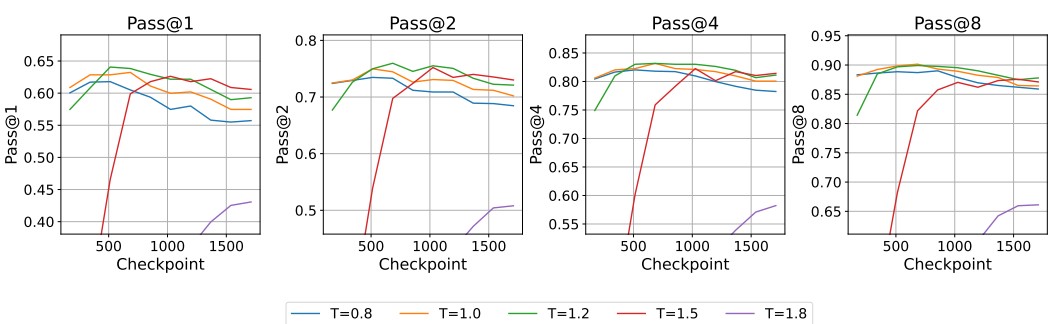

Figure 20: Pass@k of Qwen-2.5-0.5B GSM8k Oracle Top K Sampling

## F.3 Diversity Comparison Between SFT and WiSE-FT

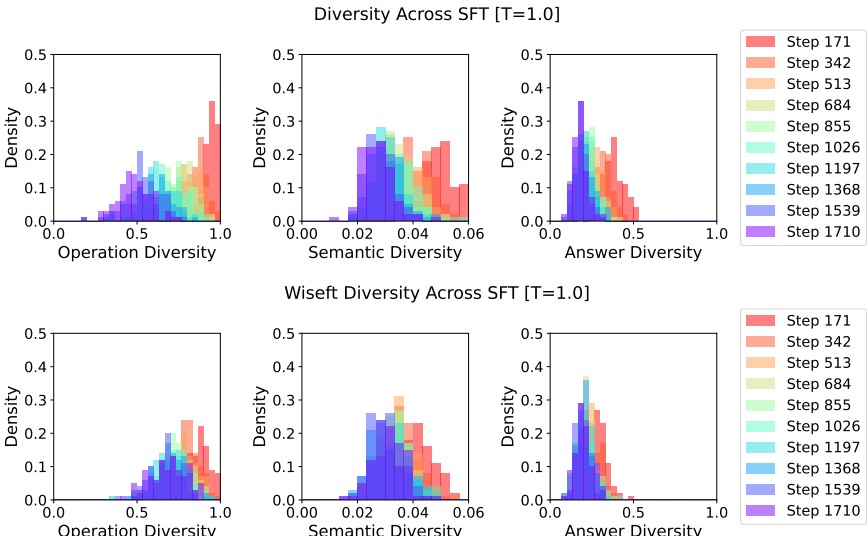

Figure 21: Operation, Semantic, and Answer Diversity of Gemma-2-2B checkpoints of SFT over GSM8K versus the corresponding WiSE-FT variants (with the earliest checkpoint). We decode with temperature set to 1.0.

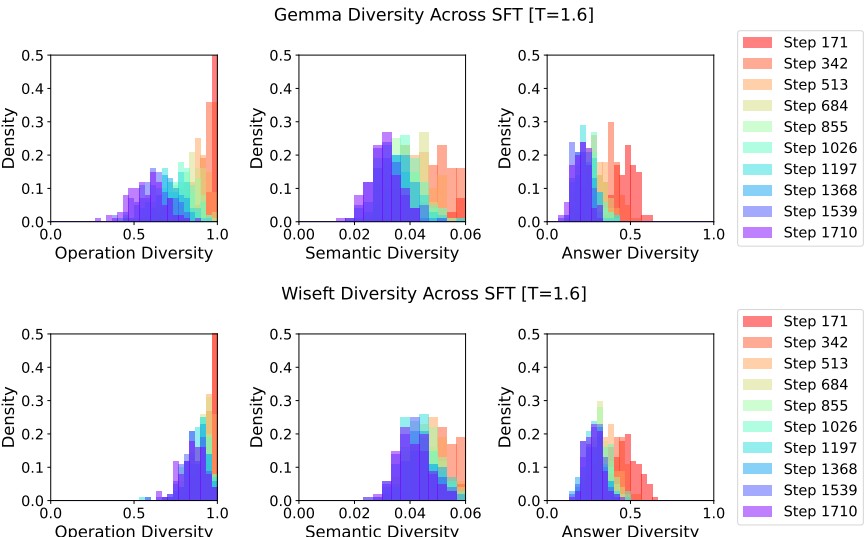

Figure 22: Operation, Semantic, and Answer Diversity of Gemma-2-2B checkpoints of SFT over GSM8K versus the corresponding WiSE-FT variants (with the earliest checkpoint). We decode with temperature set to 1.6.

# G  Best of K Evaluation

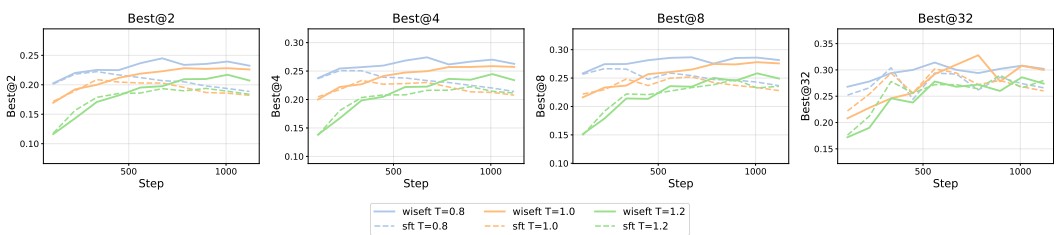

Figure 23: Best@K performance on MATH500 with ORM verifier, comparing different SFT and WiSE-FT checkpoints of Qwen-2.5-0.5B for $K = 2, 4, 8, 32$

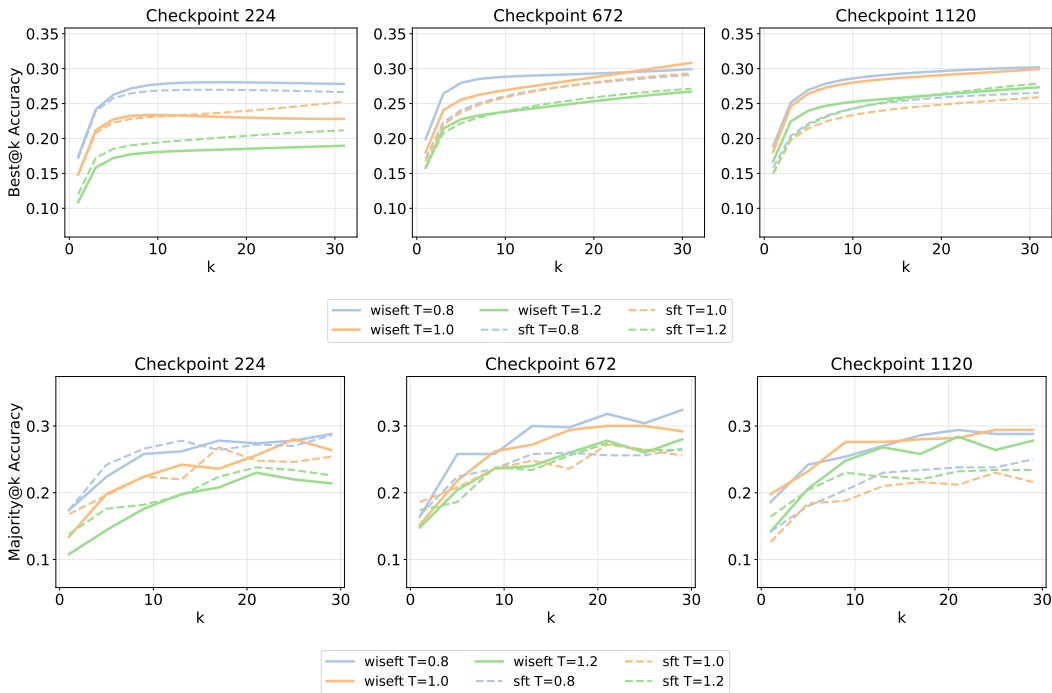

Figure 24: Best@K performance on MATH500 with ORM (Top) and Majority Vote (Bottom) for early, middle, and late SFT checkpoints and WiSE-FT counterparts, showing Qwen-2.5-0.5B 's scaling across K values.

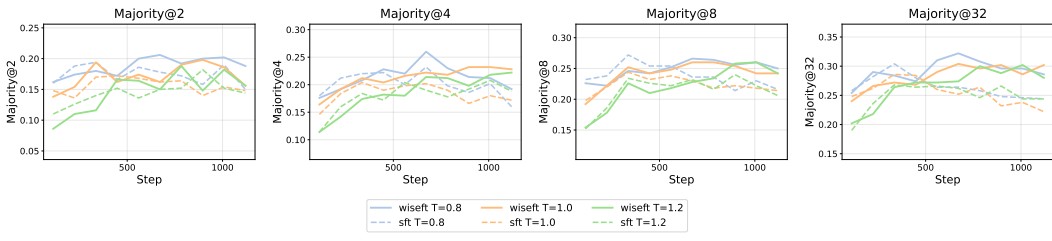

Figure 25: Best@K performance on MATH500 with majority voting, comparing different SFT and WiSE-FT checkpoints of Qwen-2.5-0.5B for $K = 2, 4, 8, 32$

# H  Diversity Collapse and WiSE-FT Results for the Coding Task

To test whether coding tasks exhibit the same diversity collapse observed in reasoning benchmarks, we fine-tuned the Qwen2.5-coder-0.5B model for 10 epochs on the Magicoder-Evol-Instruct-110K dataset, following the Stage 2 SFT recipe from OpenCoder LLM. We then applied WiSE-FT by interpolating the weights of the second SFT checkpoint with the initial model using interpolation ratio 0.5. Both the original SFT checkpoints and their WiSE-FT counterparts were evaluated on HumanEval for pass@k.

We found that, much like in mathematical reasoning tasks, SFT on coding data indeed suffers from diversity collapse: although pass@1 steadily improves over epochs, pass@k begins to deteriorate. And WiSE-FT still improves performance and mitigates the diversity collapse.

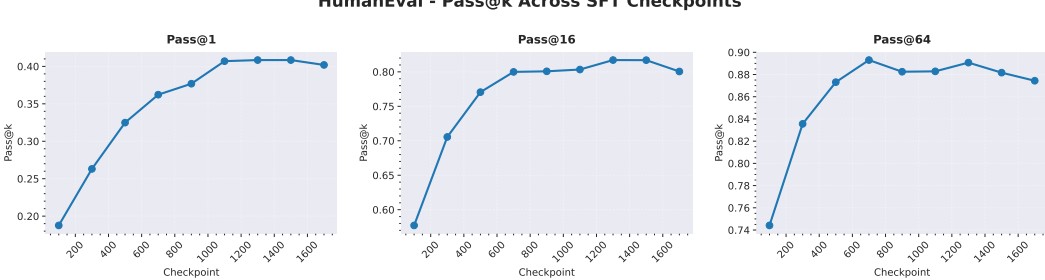

Figure 26: Pass@K performance of SFT checkpoints on HumanEval (temperature = 1.0).

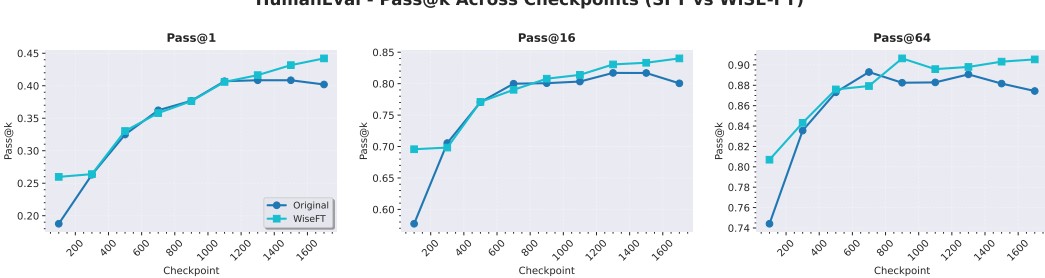

Figure 27: Comparison of pass@K for SFT checkpoints and their WiSE-FT counterparts at k = 1, 16, 64.

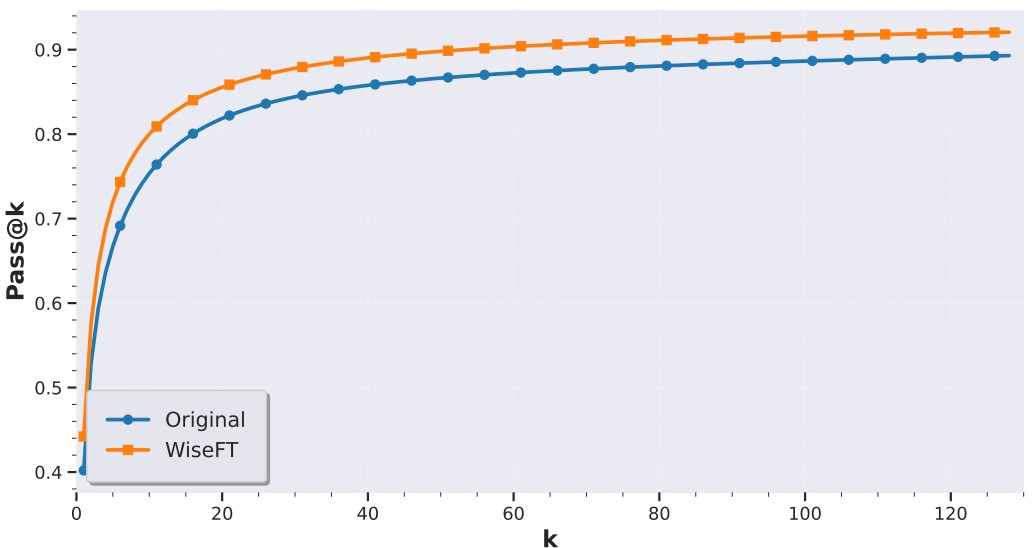

Figure 28: Pass@K performance of the final SFT checkpoint versus its WiSE-FT variant.

