# OpenReview forum: "Weight ensembling improves reasoning in language models"
_colmweb.org/COLM/2025/Conference — COLM 2025_

### Official Review · Reviewer_CtBR · 2025-05-12

**Rating:** 6
**Confidence:** 4
**Ethics Flag:** 1

**Summary:**

The paper argues that the diversity of generations collapses during supervised finetuning (SFT) for mathematical problem solving. This leads to suboptimal test-time scaling and low pass@k. The authors make use of WiSE-FT, a straightforward intervention that interpolates the weights of the latest SFT checkpoint with an early one which recovers pass@k and boosts pass@1.

The authors provide evidence for their claims across various models and datasets, complementing it with a theoretical framework that elucidates the bias-variance tradeoff in pass@k. The paper proposes a new training paradigm (SFT → WiSE-FT → RL) that offers a promising direction for developing more capable reasoning systems.

**Questions To Authors:**

Could you elaborate on the criteria used to select the "early SFT checkpoint" for WiSE-FT interpolation? Is there a principled method for determining this, or is it generally based on empirical observation?

The paper focuses on a $ 1/2 \cdot uv_0 + 1/2 \cdot uv_t$​ interpolation ratio. Have you explored other interpolation ratios for WiSE-FT, and if so, how did they affect the balance between pass@1 and pass@k, and overall performance?

Why was the Open Thoughts-114k dataset not used for the main experiments on diversity collapse, pass@k, and RL scaling, instead of the GSM8K and MATH rephrased datasets? Was there a specific reason for this choice, especially given that Open Thoughts-114k is described as a "high-quality synthetic dataset"?


Given that standard decoding strategies like nucleus and top-k fall short of oracle performance, have you investigated if WiSE-FT can be combined with these strategies to achieve synergistic effects that bring performance closer to the theoretical optimum?


In Figure 2(a), the y-axis for the "MATH Majority Vote" plot is labeled "Best@K". While "Best@K" is defined to encompass both ORM and Majority Vote, for clarity and specificity in that particular subplot, would it be more appropriate to label the y-axis as "Maj@K"?


Regarding Table 1, why are the pass@k values only shown up to k=8, given that models often demonstrate continued increases in pass@k for much larger values (e.g., K ≥ 64)? Additionally, why is the proposed WiSE-FT method not present in this table comparing various decoding strategies?

Could you elaborate on the "Top-k w/Oracle" methodology? How is the "oracle" integrated into the top-k sampling process?

Beyond the current diversity metrics (answer, operation, and semantic diversity), are there other approaches or metrics that could capture a more nuanced understanding of "meaningful" diversity in reasoning traces, especially for open-ended reasoning tasks?

**Reasons To Accept:**

The paper tackles the problem of diversity collapse in LLMs during SFT, which negatively impacts reasoning and subsequent RL training.

It proposes to use WiSE-FT, a compelling solution that is simple, practical to implement, and adds no computational overhead at inference. The method's value is demonstrated through extensive empirical results across a few models, showing consistent, and complementary gains in pass@k, pass@1, and RL scaling that are not achievable by decoding strategies alone.

The authors provide a theoretical framework that formalizes the bias-variance tradeoff, demonstrating WiSE-FT's advantages, and proposing a promising new SFT → WiSE-FT → RL training paradigm

**Reasons To Reject:**

The paper primarily focuses on a  $ 1/2 \cdot uv_0 + 1/2 \cdot uv_t$ interpolation ratio and and the use of an early SFT checkpoint. A more comprehensive exploration of different interpolation ratios and the selection criteria for the "early checkpoint" could reveal further insights into the method's generalizability and optimal application.

The analysis of the sensitivity of WiSE-FT's performance to its own hyperparameters, such as the exact choice of the early checkpoint and the interpolation weights, would strengthen the empirical results. The paper mentions using "the earliest SFT checkpoint (in our case, after 1 epoch of training)" for interpolation in WiSE-FT. However, the justification for selecting this specific checkpoint is not explicitly provided.



The theoretical analysis of diversity collapse in RL is conducted in a simplified discrete bandit setting. While insightful, the direct applicability and implications of these theoretical results to the complex, high-dimensional space of large language models could be further discussed.

The paper states that SFT is performed on a 30K subset of rephrased augmentations of GSM8k and MATH from MetaMath40k. However, the paper does not sufficiently detail the distribution of these paraphrased augmentations or elaborate on how they might contribute to the observed decay in pass@k. It would strengthen the paper to discuss whether this degradation is an inherent characteristic of finetuning on such rephrased data, or if a different source or distribution of SFT data might mitigate this issue and prevent the diversity collapse.

In the "General Reasoning Tasks" section (Section 4.3), the model is trained on "Open Thoughts-114k," a dataset of "math, science, and coding questions". However, the evaluation is only conducted on the "AIME24 competition math dataset". To demonstrate the robustness and generalizability of the results for models trained on a diverse dataset, the evaluation for this section should be extended to include a broader range of general reasoning tasks, encompassing science and coding problems, in addition to competition math.

Figure 3 provides an analysis of diversity only for SFT checkpoints. To provide a more comprehensive understanding of how WiSE-FT addresses this issue, it would be beneficial to include a similar analysis showing these diversity metrics for the WiSE-FT interpolated checkpoints.

The paper acknowledges other related work such as methods involving exploration through provers (e.g., "Rewarding Progress" by Setlur et al., 2024) and inference-aware finetuning (e.g., by Chow et al., 2024). However, a direct empirical comparison of WiSE-FT against these or similar methods that also aim to improve reasoning capabilities and diversity is missing.

The experimental evaluation primarily focuses on mathematical reasoning tasks. To strengthen the paper's claims about improving reasoning in language models more generally, it would be beneficial to extend the evaluation to other complex reasoning domains, such as code generation and scientific reasoning.

---

> ### Author Response · Authors · 2025-06-03
> **Thanks! Rebuttal Part 1**
>
> Dear Reviewer,
>
> Thank you for your very thorough feedback and recognizing the value in our work!
>
> >  **A more comprehensive exploration of different interpolation ratios**
>
> Thanks for the feedback! See Section 1 of our [rebuttal experiments](https://anonymous.4open.science/r/COLM2025_Rebuttal-EA87/COLM_2025_Rebuttal_Figures.pdf) for the ablation study over the choice of interpolation coefficients. Generally, we find that 0.5 yields good performance across multiple tasks (GSM8k, MATH500, AIME24) and model architectures. Therefore, we choose 0.5 uniformly for all tasks experimented in our paper. Furthermore, we see the same smooth wishbone-shaped curve from the original WiSE-FT work, where there is a regime where Pass@1 and Pass@K are both higher than the early and later SFT models.
>
> > **Could you elaborate on the criteria used to select the "early SFT checkpoint" for WiSE-FT interpolation?**
>
> Thanks for the great question! First, we agree with you that an ablation study that performs WiSE-FT with different checkpoints could be informative. We’ve added this result in Section 3 of our [rebuttal experiments](https://anonymous.4open.science/r/COLM2025_Rebuttal-EA87/COLM_2025_Rebuttal_Figures.pdf), and we’ll also add this to the camera ready if accepted.
>
> To build more intuition, a main takeaway from our paper is that SFT (+RL)  faces a detrimental _tradeoff_ between Pass@1 and Pass@K. Due to diversity collapse, Pass@K begins to drop although Pass@1 continues to improve.
> When two models have a tradeoff between metrics, WiSE-FT can surprisingly achieve “best of both world” gains ([Wortsman et al](https://arxiv.org/abs/2109.01903)).  We apply this idea to the Pass@1 vs Pass@K tradeoff. We take
> 1. Model A with higher Pass@K but low Pass@1
> 2. Model B with lower Pass@K but higher Pass@1
> 3. WiSEFT between Model A and Model B  to get both high Pass@K and high Pass@1.
>
> We choose the earliest checkpoint to serve as Model A because Pass@K for higher values of K tends to strictly drop for later SFT checkpoints (see Pass@32 in Figure 1). On the other hand, since Pass@1 continues to improve in our SFT runs,  we choose the last checkpoint for Model B. We avoid the base model as the choice of Model A, because some light finetuning is needed to teach the model the reasoning structure and allow us to extract answers reliably from the model for evaluation.
> Currently, this is discussed briefly in the preliminary section in Line 133, but we will revise our results section to discuss this more explicitly.
>
> > **Theoretical analysis of RL on bandit problem may be too simple to explain the real-world phenomena**
>
> Thank you for finding our bandit theory insightful. We would like to clarify that this simplified setting is primarily intended to build intuition rather than to faithfully model the exact dynamics of high-dimensional neural networks, which are far more complex and beyond the scope of this paper. We will make this intention clearer in the revised version.
>
> > **How might the paraphrasing of GSM8K and MATH in MetaMath40k contribute to the observed decay in Pass@K?**
>
> We apologize for any confusion — both our theoretical and empirical analyses indicate that diversity collapse is a result of _the fundamental overconfidence issue present in large models_ that overfit to finite fine-tuning datasets. We explore this in detail in Sections 5 and 6. This overconfidence issue appears even in simple models with linearly separable data (see Appendix B).
>
> Diversity collapse is _not at all_ a direct result of using MetaMath instead of the original GSM8K/MATH datasets. In fact, we confirm that the drop in Pass@K is far worse when training on the original GSM8K and MATH datasets, because the dataset is smaller and they are easier to overfit. Generally, increasing the size of the dataset by synthetic data augmentation should help with the overconfidence issue.
>
> > **Only focus on Math tasks**
>
> We believe the phenomenon is general and are currently conducting experiments on the code generation task, we’ll get back later for the results of this experiment.
>
> > **Do diversity analysis on WiSE-FT outputs as well, to compare against SFT**
>
> Thank you for your feedback! See Section 2 of our  [rebuttal experiments](https://anonymous.4open.science/r/COLM2025_Rebuttal-EA87/COLM_2025_Rebuttal_Figures.pdf) for this ablation study. The WiSE-FT histograms see a consistent shift to the right for all diversity types, as the outputs are generally more diverse.  We’ll incorporate this result in the camera ready.

---

> ### Author Response · Authors · 2025-06-03
> **Rebuttal Part 2**
>
> > **Need comparison with other diversity-preserving methods. Can you show synergistic benefits if you apply WiSE-FT with diverse decoding strategies (min-p, top-K, nucleus)?**
>
> Great question! WiSE‑FT is intended as a lightweight, complementary tool to address diversity collapse. Any decoding‑time diversity‑preserving method can be applied on top of the weight‑ensembled model, and we do not claim that WiSE‑FT alone strictly outperforms those methods. Therefore, we will omit direct baseline comparisons.
>
> Decoding strategies and WiSE-FT can be tuned to have synergistic benefits. In our paper, we’ve generally fixed the decoding temperature for the WiSE-FT model to T=1.0. But we’ve seen that by also tuning the temperature parameter, you can achieve an even higher Pass@1 or Pass@K than what we report. We leave a more careful exploration of effective combinations of WiSE‑FT with decoding strategies for future work!
>
> > **What's the “Top-k w/Oracle” methodology? How is the "oracle" integrated into the top-k sampling process?**
>
> For each question, we estimate the marginal probability of each candidate answer by averaging over 1,000 independent rollouts (e.g., a model’s reasoning trace may converge to answer A x% of the time, answer B y% of the time, etc.). We then select the top K answers according to these marginal probabilities. In effect, this simulates an optimal decoding strategy of top-k sampling assuming the marginal answer distribution is known.
>
> > **Regarding Table 1, why are the Pass@k values only shown up to k=8?...[add] WiSE-FT method [to] this table comparing various decoding strategies**
>
> Great question! To clarify, Table 1 depicts the _gap_ between current diversity-preserving strategies and the optimal Pass@K given oracle access to the marginal answer distribution. Surprisingly, the gap is already significant even for small K=2,4,8, meaning current strategies aren’t able to sample rollouts with diverse answers efficiently. The Table is not intended to show how Pass@K tracks for all decoding strategies for all K.
>
> Like other strategies, WiSE-FT also falls below this optimal upper bound and we don’t intend to suggest otherwise! As you’ve suggested, we’ve modified the table to include WiSE-FT numbers in Section 4 of our  [rebuttal experiments](https://anonymous.4open.science/r/COLM2025_Rebuttal-EA87/COLM_2025_Rebuttal_Figures.pdf). We also noticed that the caption in our submission states that the model for this study is Gemma, but the model here is actually Qwen. We’ll update the Table 1 with these numbers and caption correction in our camera ready.
>
> > **ORM and Majority Vote, for clarity and specificity in that particular subplot, would it be more appropriate to label the y-axis as "Maj@K"?**
>
> Thank you for catching this! We agree that using “Maj@K” would improve clarity. We will update the figure and its description accordingly.
>
> Hope this clarified your questions! Please let us know if you have any other feedback for us!
>
> Authors

---

> ### Author Response · Authors · 2025-06-08
> **Update of Experiments on Coding Task**
>
> > **Only focus on Math tasks**
>
> Thank you for pointing out that we should verify the diversity collapse phenomenon on coding. We have now added a new ablation on the coding task in our [rebuttal experiments](https://anonymous.4open.science/r/COLM2025_Rebuttal-57B6/) (Section 5). Specifically, we fine-tuned the Qwen2.5-coder-0.5B model on Magicoder-Evol-Instruct-110K for 10 epochs, then applied WiSE-FT by interpolating the second SFT checkpoint with the base model (ratio=0.5). Both SFT and WiSE-FT checkpoints were evaluated on HumanEval via pass@k.
>
> We observe that coding data indeed exhibits the same diversity collapse as in reasoning tasks—pass@1 improves while pass@K degrades, and larger K degrades earlier —and that WiSE-FT both boosts overall pass@1 and restores pass@K diversity.

---

> ### Author Response · Authors · 2025-06-10
> **Checking in**
>
> Hi Reviewer CtBR,
>
> Did our rebuttal experiments fully address your questions? If you any other remaining concerns, we're happy to address them. Please let us know!
>
> Best,
> Authors

---

### Official Review · Reviewer_Mseo · 2025-05-13

**Rating:** 7
**Confidence:** 3
**Ethics Flag:** 1

**Summary:**

This paper proposes the usage of a simple weight ensembling technique, WiSE-FT, to address the diversity collapse problem that happens after model finetuning, either SFT or RL. WiSE-FT simply proposes to mix the finetuned model weights with vanilla model weights evenly, and improves the pass@k for the fintuned model. The paper further discusses some theoretical foundations of the diversity collapse issue, and how it impacts pass@k, and further test-time scaling. The proposed method and theoretical conjectures are evaluated empirically by finetuning Gemma and Qwen models with SFT and RL.

**Questions To Authors:**

- Line 243: It would be interesting to see if the KL loss can help SFT.

**Reasons To Accept:**

- The proposed method is simple and yet effective. It's more likely to be adapted in various finetuning recipes.
- I like the theoretical analyses. They give more insights into how the diversity collapse problem impacts the widely used recipe of SFT followed by RL. These analyses are well supported by numbers and experiments.

**Reasons To Reject:**

- The impact of finetuning RL is only backed by theory, but no empirical evidence.
- It's not clear if the pass@1 improvement is valid. There seems to be no clear theoretical support.

---

> ### Author Response · Authors · 2025-06-02
> **Thanks**
>
> Dear Reviewer,
>
> Thank you for your feedback and recognizing the value in our work!
>
> >**“The impact of finetuning RL is only backed by theory, but no empirical evidence.”**
>
> There could be a misunderstanding here. We do provide empirical evidence that we think you’re asking for in Figure 2C and 4. In Figure 4, we show that RL _is not able to recover from the drop in Pass@K_. The purple solid line measures Pass@K across SFT steps, while the three dashed lines correspond to Proximal Policy Optimization (PPO) starting from different SFT checkpoints. For very large K, we see that the Pass@K performance continues to drop with longer RL. For medium K, we observe RL improves Pass@K, but its not able to fully recover the drop from SFT.  Please let us know if this answers your question or if you were looking for another kind of RL experiment.
>
> > **“It's not clear if the pass@1 improvement is valid.  There seems to be no clear theoretical support.”**
>
> Thanks for the feedback! The observation we make that SFT continues to improve Pass@1 is only an empirical. Our paper does not attempt to explain why Pass@1 improves during SFT.  We also don’t intend to argue that Pass@1 will always improve. There may be an overfitting regime if we were to train for even longer. Our theory focuses on why _Pass@K often drops faster than Pass@1_, which we go into detail in Sections 5 and 6 of our paper. Several works have also noted this phenomenon previously and we discuss them in our Related Works (Lines 82-96). If you think there is a particular place we overclaim, please let us know.
>
> You might also be interested in our [additional ablation study](https://anonymous.4open.science/r/COLM2025_Rebuttal-EA87/COLM_2025_Rebuttal_Figures.pdf) where we interpolated the early and later SFT model across interpolation coefficients and plotted the Pass@1 versus Pass@K. We see the same smooth wishbone-shaped curve from the original WiSE-FT work, where there is a regime where Pass@1 and Pass@K are both higher than the early and later SFT models.
>
> Does this address your concerns? If you have any other feedback for improving our work, please let us know!
>
> Authors

---

> > ### Comment · Reviewer_Mseo · 2025-06-09
> >
> > Thanks for the clarification!
> >
> > re: my question of finetuning RL: Please correct if my understanding is wrong, but it seems Figure 4 is more like an indirect support for diversity collapse during RL finetuning. Does it make sense to show some more direct evidences like Figure 3?

---

> > ### Author Response · Authors · 2025-06-10
> > **Diveristy Histograms for PPO**
> >
> > This is a great suggestion, thank you.
> >
> > We've conducted the experiment you asked for in Section 7 of our [rebuttal experiments](https://anonymous.4open.science/r/COLM2025_Rebuttal-EA87/COLM_2025_Rebuttal_Figures.pdf).
> > We've measured diversity collapse across RL steps over GSM8k. These are the same RL checkpoints we've evaluated in Figure 3.  Similar to SFT, we observe a clear decrease in diversity with more steps of PPO --- the histograms strictly shift to the left.
> >
> > We'll add this to the camera ready of the paper if accepted. Let us know if this answers your questions. If you have any other remaining concerns, we're happy to address them!

---

> > > ### Comment · Reviewer_Mseo · 2025-06-10
> > >
> > > Great. Thanks for the additional information!

---

### Official Review · Reviewer_yBuj · 2025-05-13

**Rating:** 7
**Confidence:** 3
**Ethics Flag:** 1

**Summary:**

Prior work has found that during supervised finetuning of reasoning models, the single best generation (pass@1) can continue to improve, while the best of K generations (pass@K) begins to deteriorate. This is due to diversity collapse, as SFT-ed models tend to converge to a single reasoning trace.

This paper shows that by simply interpolating the weights of the latest SFT checkpoint with an early checkpoint (a method called WiSE-FT from Wortsman et al., 2022), the pass@K performance can be completely recovered while also improving pass@1. Due to preserving output diversity, there are two immediate benefits of the WiSE-FT model: it scales better with test-time compute (§4.1), and is a better starting point for RLHF (§4.2). These gains cannot be achieved using an early-stopped SFT model or high-temperature decoding from the final SFT checkpoint. The experiments mainly use math datasets like GSM8K and MATH, though they show their results also generalize to training on OpenThoughts-114k.

The authors show empirically that diversity collapse happens with more steps in SFT (§5.1), and then connect this to pass@K in some theoretical discussion (§6). They prove that the expected pass@K decreases with increased variance in pass@1. SFT causes models to become very confident, so that the pass@1 distribution becomes bimodal — it is either very low (when the model is wrong) or very high (when the model is right). This increase in variance explains the drop in pass@K.

**Questions To Authors:**

Notes on presentation:
- Figure 2 caption is hard to parse due to formatting.
- In Figure 2c, the color-coding of "SFT Early" and "SFT Last" switches between the top and bottom subfigure.
- L. 252: "diveristy" $\rightarrow$ "diversity"

**Reasons To Accept:**

1. The paper is very well-written and has an insightful discussion of the tradeoff between pass@1 and pass@K during SFT.
2. The method, WiSE-FT, is extremely effective. It successfully improves *both* pass@1 and pass@K with longer SFT. The resulting models scale better with increased test-time compute, and are substantially better starting points for RLHF.
3. There is meaningful analysis & theoretical discussion of how SFT leads to diversity collapse and compromises pass@K.

**Reasons To Reject:**

1. This is not a reason to reject in my opinion, but it is worth noting that the method is not new. In the original paper, WiSE-FT interpolates the weights of a finetuned model and pretrained model to improve OOD generalization. The authors give credit to the original work and do not attempt to downplay the connection.
2. In the related work, it would be good to discuss other successful applications of model souping for LLMs. For instance, I know that [Olmo 2](https://arxiv.org/abs/2501.00656) souped model checkpoints trained on different data permutations, which they found to match or outperform any individual checkpoint.

---

> ### Author Response · Authors · 2025-06-01
> **Thanks**
>
> Dear Reviewer,
>
> Thank you for your feedback and recognizing the value in our work!
>
> > **“good to discuss other successful applications of model souping for LLMs”**
>
> Thanks for the suggestion! We’ve also seen successful uses of model soup in LLMs for
> 1. Ensembling LLMs fine-tuned on different tasks ([Llama2](https://arxiv.org/abs/2307.09288), [Ilharco et al.](https://arxiv.org/abs/2212.04089). [OLMO2](https://arxiv.org/abs/2501.00656))
> 2. LLMs pretrained on different specialized domains ([Ablin et al.](https://arxiv.org/abs/2502.01804))
> 3. Continual learning algorithms tend to incorporate similar ideas ([CoMA](https://arxiv.org/abs/2312.08977), [EWC](https://arxiv.org/pdf/1612.00796)).
>
> We’ll add this discussion of papers to our related works. If you are  aware of other papers, please let us know.
>
> > **Figure 2 caption is hard to parse due to formatting**
>
> Thank you for catching the coloring and spacing issues. We’ll work on shortening the Figure 2 caption and make the plot color consistent across subplots for the revised version.
>
> You might also be interested in our [additional ablation study](https://anonymous.4open.science/r/COLM2025_Rebuttal-EA87/COLM_2025_Rebuttal_Figures.pdf) where we interpolated the early and later SFT model across interpolation coefficients and plotted the Pass@1 versus Pass@K. We see the same smooth wishbone-shaped curve from the original WiSE-FT work, where there is a regime where Pass@1 and Pass@K are both higher than the early and later SFT models.
>
> If you have any other suggestions for improving our work, please let us know!
>
> Authors

---

> > ### Comment · Reviewer_yBuj · 2025-06-03
> >
> > Thanks for sharing the additional results!

---

### Official Review · Reviewer_jb4c · 2025-05-14

**Rating:** 8
**Confidence:** 4
**Ethics Flag:** 1

**Summary:**

This paper focuses on balancing Pass@1 and Pass@k for reasoning models. The authors present a finding that supervised fine-tuning (SFT) improves Pass@1 but reduces output diversity. Consequently, the fine-tuned model is not well-suited for subsequent reinforcement learning alignment. Based on this finding, the authors propose a weight ensemble method, which averages the weights from different stages of SFT. Surprisingly, this simple ensemble performs well on both Pass@1 and Pass@k, even outperforming early stopping in SFT. The authors present extensive results to support their conclusions.

**Questions To Authors:**

Please see above.

**Reasons To Accept:**

- The paper presents solid evidence showing how SFT affects the performance of Pass@1 and Pass@k, as well as output diversity.
- The proposed ensemble method is simple but highly effective.
- The authors provide theoretical explanations for their findings.
- Extensive experimental results are presented.

**Reasons To Reject:**

I just have one minor question: what optimizer is used in the experiments? Given that the ensemble method performs better than early-stopped SFT, I assume part of the reason is that the gradient direction changes significantly over time. If the gradient direction is always similar during the optimization, there should not be a significant different between ensemble and early stopping. I'm wondering whether the choice of optimizer could significantly affect this conclusion. It's worth trying other optimizer for a better insight.

Other than that, I don't see any major weaknesses.

---

> ### Author Response · Authors · 2025-06-01
> **Thanks**
>
> Dear Reviewer,
>
> Thank you for your feedback and recognizing the value in our work!
>
> You make a really great point about how weight ensembling works because the gradient trajectory is nonlinear.  For the optimizer, we used the standard AdamW with learning rate ranging between 1e-4 to 1e-5.  Also, we do SFT for thousands of steps in all our experiments. You’re right that if the gradient direction did not change, we would not see this “best-of-both worlds” effect compared to early-stopping. In other words, the model is less overconfident than the final SFT checkpoint but retains higher Pass@1 than the early stopped checkpoint.
>
> Great point, we will definitely include an ablation study comparing AdamW with other optimizers in the camera ready. We also refer you to the original WiSE-FT paper “Robust fine-tuning of zero-shot models” where in Figure 3, they conducted an ablation study with different learning rates, epochs, and optimizers (Adam, AdamW, and SGD). They were able to see the same “best of both worlds” effect across all these settings.
>
> You might also be interested in our [additional ablation study](https://anonymous.4open.science/r/COLM2025_Rebuttal-EA87/COLM_2025_Rebuttal_Figures.pdf) where we interpolated the early and later SFT model across interpolation coefficients and plotted the Pass@1 versus Pass@K. We see the same smooth wishbone-shaped curve from the original WiSE-FT work, where there is a regime where Pass@1 and Pass@K are both higher than the early and later SFT models.
>
> Please feel free to raise other questions or concerns if you have any!
>
> Authors

---

> > ### Comment · Reviewer_jb4c · 2025-06-02
> >
> > Thanks for the reply. I am looking forward to the ablation study.

---

> > > ### Author Response · Authors · 2025-06-09
> > > **Update on optimizer ablation experiments**
> > >
> > > We thank the reviewer for emphasizing the potential impact of optimizer choice on our conclusions. As requested, we have carried out an ablation study comparing AdamW against Momentum SGD when fine-tuning Qwen2.5-0.5B on the MATH split of MetaMath and evaluating on MATH500. We tuned the hyper-params of Momentum SGD but we still observed a substantial performance gap between the two optimizers. Please refer to section 6 of our [updated ablation studies](https://anonymous.4open.science/r/COLM2025_Rebuttal-57B6/COLM_2025_Rebuttal_Figures.pdf)
> > >
> > > This finding is consistent with well-established observations in the previous works that Adam’s adaptive coordinate‐wise preconditioning and gradient‐clipping mitigate the sharp‐minima and heavy‐tailed stochastic noise that typically impair SGD on LLMs.
> > >
> > > (Pan & Li, 2023, https://arxiv.org/abs/2306.00204)
> > >
> > > (Zhang et al. 2019, https://arxiv.org/abs/1912.03194)
> > >
> > > (Zhao et al. 2024, https://arxiv.org/abs/2407.07972v1)

---

### Decision · Program_Chairs · 2025-07-06

**Decision:**

Accept

**Comment:**

The paper proposes a technique to improve the diversity of generations of a model which is post-trained using either RL or SFT, by a simple weight ensembling technique. Reviewers generally agree that the approach is simple but highly effective (jb4c, yBuj, CtBR, Mseo) and can be applicable to improve post-trained model's diversity in a general setting. Reviewers are also satisfied with the theoretical justifications provided in the paper. CtBR had several questions and comments regarding the empirical methods in the paper, which the authors respond adequately. Given the positive reviews, I'm also happy to recommend an accept.